# HG-Adapter: Improving Pre-Trained Heterogeneous Graph Neural Networks with Dual Adapters

**Yujie Mo**[1,2]  **Runpeng Yu**[2]  **Xiaofeng Zhu**[1*]  **Xinchao Wang**[2*]

[1]School of Computer Science and Engineering,
University of Electronic Science and Technology of China
[2]National University of Singapore

## Abstract

The "pre-train, prompt-tuning" paradigm has demonstrated impressive performance for tuning pre-trained heterogeneous graph neural networks (HGNNs) by mitigating the gap between pre-trained models and downstream tasks. However, most prompt-tuning-based works may face at least two limitations: (i) the model may be insufficient to fit the graph structures well as they are generally ignored in the prompt-tuning stage, increasing the training error to decrease the generalization ability; and (ii) the model may suffer from the limited labeled data during the prompt-tuning stage, leading to a large generalization gap between the training error and the test error to further affect the model generalization. To alleviate the above limitations, we first derive the generalization error bound for existing prompt-tuning-based methods, and then propose a unified framework that combines two new adapters with potential labeled data extension to improve the generalization of pre-trained HGNN models. Specifically, we design dual structure-aware adapters to adaptively fit task-related homogeneous and heterogeneous structural information. We further design a label-propagated contrastive loss and two self-supervised losses to optimize dual adapters and incorporate unlabeled nodes as potential labeled data. Theoretical analysis indicates that the proposed method achieves a lower generalization error bound than existing methods, thus obtaining superior generalization ability. Comprehensive experiments demonstrate the effectiveness and generalization of the proposed method on different downstream tasks.

## 1 Introduction

Pre-trained heterogeneous graph neural networks (HGNNs) are designed to pre-train models on the heterogeneous graph data and then effectively generalize to diverse tasks (Fan et al., 2019; Jiang et al., 2021). To achieve this, current pre-trained HGNNs typically utilize unsupervised techniques during pre-training to learn fundamental properties, thereby enhancing the generalization ability of models (Yang et al., 2022; Fan et al., 2024). Consequently, pre-trained HGNNs have demonstrated promising potential in real applications such as recommendation systems, social network analysis, and molecular design (Shi et al., 2016; Tian et al., 2023; Wu et al., 2024).

Existing pre-trained HGNNs generally follow two paradigms, *i.e.,* "pre-train, fine-tuning" and "pre-train, prompt-tuning". The "pre-train, fine-tuning" paradigm typically first trains the model with unlabeled data in the pre-training stage, and then updates the pre-trained model with task-related labels in the fine-tuning stage to adapt it to downstream tasks (Wang et al., 2021; Tian et al., 2023). However, the two stages in the "pre-train, fine-tuning" paradigm optimize different objectives, resulting in the gap between pre-trained models and downstream tasks that weakens the model generalization (Liu et al., 2023a; Yu et al., 2024c). To solve this issue, recent works propose the "pre-train, prompt-tuning" paradigm to connect pre-trained models with downstream tasks by designing a learnable prompt that directly appends to (or modifies) the model input (Yu et al., 2024b; Liu et al., 2023b). For

---

*Corresponding authors

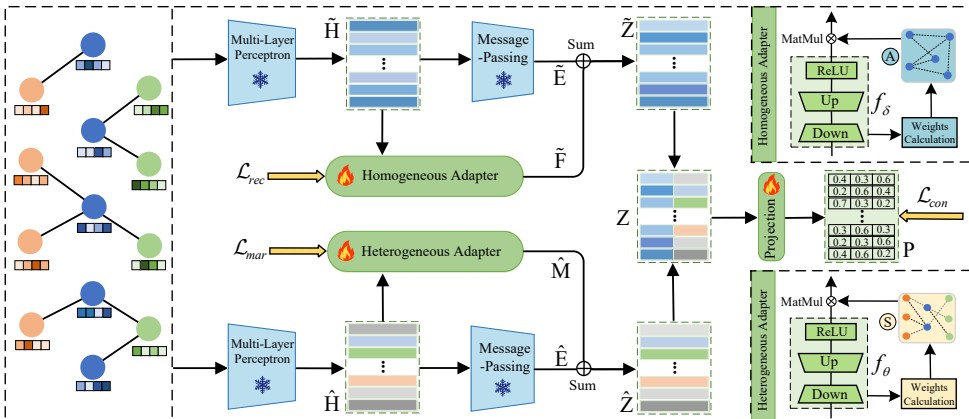

Figure 1: The flowchart of the proposed HG-Adapter. Given the frozen representations $\tilde{\mathbf{H}}$ from the pre-trained model, the homogeneous adapter is designed to generate the adapted representations $\tilde{\mathbf{F}}$ by tuning the homogeneous graph structure $\mathbf{A}$ (details in the top right corner). After that, the homogeneous representations $\tilde{\mathbf{Z}}$ are obtained by summing the frozen representations $\tilde{\mathbf{E}}$ after message-passing and $\tilde{\mathbf{F}}$. Similarly, the heterogeneous representations $\hat{\mathbf{Z}}$ are obtained by summing the frozen representations $\hat{\mathbf{E}}$ after message-passing and the adapted representations $\hat{\mathbf{M}}$ from the heterogeneous adapter (details in the bottom right corner). Furthermore, $\tilde{\mathbf{Z}}$ and $\hat{\mathbf{Z}}$ are concatenated to generate $\mathbf{Z}$, which is then mapped to the prediction matrix $\mathbf{P}$. Finally, HG-Adapter designs a label-propagated contrastive loss (i.e., $\mathcal{L}_{con}$) and two self-supervised losses (i.e., $\mathcal{L}_{rec}$ and $\mathcal{L}_{mar}$) to optimize dual adapters and extend potential labeled data to improve the model generalization.

instance, HGPrompt (Yu et al., 2024a) proposes dual-template and dual-prompt to unify pre-training and downstream tasks for both homogeneous and heterogeneous graphs. Similarly, HetGPT (Ma et al., 2024) introduces virtual class and heterogeneous feature prompts, and then reformulates the downstream objective function to align it with pre-training tasks.

However, existing "pre-train, prompt-tuning" methods have at least two limitations. First, the prompt-tuning in existing methods only focuses on the node features while ignoring the graph structures in the heterogeneous graph. As a result, the model may not sufficiently fit the task-related information in graph structures during the prompt-tuning stage, thereby increasing the training error and decreasing the generalization ability on downstream tasks (Bousquet & Elisseeff, 2002). Second, the pre-trained models are generally trained in an unsupervised manner, thus their ability to generalize to unlabeled data may be constrained by the limited labeled data during the prompt-tuning stage. Consequently, this may result in a large generalization gap between the training error and the test error, leading to sub-optimal generalization performance (Arora et al., 2018).

To alleviate the above limitations and improve the model generalization, three challenges remain to be addressed. (i) Existing prompt-tuning-based methods usually rely on heuristically designed prompts to improve generalization but lack a unified theoretical framework, highlighting the necessity to formally understand and derive the key factors that affect the generalization of existing methods. (ii) Based on such theoretical foundation, a new framework needs to be designed to effectively capture the task-related structural information in the heterogeneous graph, thus decreasing the training error and improving the model generalization. (iii) Given the difficulty of directly increasing the labeled data, new alternative solutions are required to mitigate the issue of limited task-related labels in the tuning stage, thereby decreasing the generalization gap and further improving the model generalization.

In this paper, we address the above challenges by first deriving a generalization error bound for existing prompt-tuning-based methods and then proposing a novel adapter-tuning-based framework (termed HG-Adapter) with dual structure-aware adapters and potential labeled data extension, thus improving the generalization of pre-trained HGNNs, as shown in Figure 1. Specifically, we first theoretically analyze the generalization error bound for existing methods based on the training error and the generalization gap, thus building the theoretical foundation for our method and exploring **challenge (i)**. After that, we design homogeneous and heterogeneous adapters to capture the task-related

structural information by adaptively tuning both homogeneous and heterogeneous graph structures, thus decreasing the training error and improving the model generalization to explore **challenge (ii)**. Moreover, we design a label-propagated contrastive loss and two self-supervised losses on unlabeled nodes, thus potentially extending the labeled data and decreasing the generalization gap to further improve the model generalization and explore **challenge (iii)**. Finally, we theoretically demonstrate that the proposed method achieves a lower generalization error bound than existing prompt-tuning-based methods, leading to superior performance across different downstream tasks.

Compared to existing works, our main contributions are summarized as follows:

- To our best knowledge, this is the first dedicated attempt to design a unified "pre-train, adapter-tuning" paradigm to improve different pre-trained HGNN models.

- We design dual structure-aware adapters to capture task-related homogeneous and heterogeneous structural information. Moreover, we design a label-propagated contrastive loss and two self-supervised losses to achieve the potential labeled data extension.

- We theoretically derive a unified generalization error bound for existing methods based on the training error and the generalization gap. Moreover, we demonstrate that the proposed method achieves a lower generalization error bound than existing prompt-tuning-based methods to improve the generalization ability of pre-trained HGNN models.

- We experimentally validate the superior effectiveness and generalization of the proposed HG-Adapter compared to state-of-the-art fine-tuning-based and prompt-tuning-based methods, and demonstrating its adaptability to different pre-trained HGNN models.

## 2 METHOD

### 2.1 PRELIMINARIES

**Definition 2.1.** *Heterogeneous graph (Sun & Han, 2012) is defined as $\mathbf{G} = (\mathcal{V}, \mathcal{E}, \mathcal{A}, \mathcal{R}, \phi, \varphi)$, where $\mathcal{V}$ and $\mathcal{E}$ indicate the set of nodes and the set of edges, respectively. $\mathcal{A}$ and $\mathcal{R}$ indicate the set of node types and the set of edge types, respectively, where $|\mathcal{A} \cup \mathcal{R}| > 2$ and the node type relevant to downstream tasks is referred to as the target node. Moreover, $\mathcal{A}$ and $\mathcal{R}$ are associated with the node type mapping function $\phi : \mathcal{V} \rightarrow \mathcal{A}$ and the edge type mapping function $\varphi : \mathcal{E} \rightarrow \mathcal{R}$, respectively.*

**Definition 2.2.** *Pre-trained HGNNs (Wang et al., 2021; Mo et al., 2024) generally include homogeneous and heterogeneous branches, which can be divided into two steps: 1) mapping the original node features $\mathbf{X}$ with the Multi-Layer Perceptron (MLP) to obtain low-dimensional representations, and 2) conducting the message-passing among nodes from the same type and nodes from different types based on homogeneous and heterogeneous graph structures, respectively, i.e.,*

$$\tilde{\mathbf{H}} = \text{MLP}(\mathbf{X}), \tilde{\mathbf{E}} = \text{Message-Passing}(\tilde{\mathbf{H}}, \mathcal{G}_{hom}),$$
$$\hat{\mathbf{H}} = \text{MLP}(\mathbf{X}), \hat{\mathbf{E}} = \text{Message-Passing}(\hat{\mathbf{H}}, \mathcal{G}_{het}),$$

(1)

*where $\mathcal{G}_{hom}$ indicates the homogeneous graph structure that connects nodes within the same type, and $\mathcal{G}_{het}$ indicates the heterogeneous graph structure that connects nodes from different types. The homogeneous and heterogeneous graph structures generally come from pre-defined meta-paths and the given set of edges $\mathcal{E}$ in the heterogeneous graph, respectively.*

### 2.2 GENERALIZATION BOUND OF PROMPT-TUNING-BASED METHODS

Given pre-trained HGNN models, existing "pre-train, prompt-tuning" methods typically freeze the model parameters and design tunable prompts to modify the input and capture task-related information (Yu et al., 2024a). Despite effectiveness, existing methods generally rely on heuristically designed prompts and lack a unified theoretical framework to further improve the generalization ability. As a result, during the prompt-tuning stage, most existing methods ignore graph structures and may suffer from limited labeled data, leading to inferior model generalization. To address this issue, we derive the generalization error bound for existing prompt-tuning-based methods (*i.e.,* HGPrompt (Yu et al., 2024a) and HetGPT (Ma et al., 2024)) based on the classical regime of the generalization bound theory (Arora et al., 2018; Aghajanyan et al., 2021) (proofs are provided in Appendix C.1).

**Theorem 2.3.** *(Generalization error bound for prompt-tuning-based methods.) Statistically, the upper bound $\mathcal{U}(\mathcal{E}_M)$ of the test error $\mathcal{E}_M$ of a pre-trained HGNN model with prompt-tuning can be determined as follows:*

$$\mathcal{U}(\mathcal{E}_M) = \hat{\mathcal{E}}_M(\mathcal{D}_{n_M}, \mathcal{P}_M) + O(\sqrt{|\mathcal{P}_M|/n_M}), \tag{2}$$

*where training data $\mathcal{D}_{n_M}$ and prompt-tuning parameters $\overline{\mathcal{P}}_M$ are variables of the training error $\hat{\mathcal{E}}_M$ of the model in prompt-tuning stage. The number of training samples $n_M$ and the size of parameter space $|\mathcal{P}_M|$ are variables of the generalization gap bound between training error and test error. Moreover, when $n_M$ is fixed, there exist an optimal $|\overline{\mathcal{P}}_M|$ to achieve the lowest upper bound for prompt-tuning-based methods, i.e.,*

$$\min(\mathcal{U}(\mathcal{E}_M)) = \hat{\mathcal{E}}_M(\mathcal{D}_{n_M}, \overline{\mathcal{P}}_M) + O(\sqrt{|\overline{\mathcal{P}}_M|/n_M}). \tag{3}$$

Theorem 2.3 indicates that the lowest generalization error bound of existing prompt-tuning-based methods exists and consists of two parts, *i.e.,* the training error $\hat{\mathcal{E}}_M$ of the model in the prompt-tuning stage, and the generalization gap bound $O(\sqrt{|\overline{\mathcal{P}}_M|/n_M})$. As a result, we theoretically derive the generalization error bound of existing prompt-tuning-based methods based on the training error and the generalization gap, thus solving the challenge (i).

## 2.3 DUAL STRUCTURE-AWARE ADAPTERS

Intuitively, based on Theorem 2.3, when $n_M$ is fixed, encouraging the parameters $\mathcal{P}_M$ to approach the optimal parameters $\overline{\mathcal{P}}_M$ can enable the model to approach the lowest generalization error bound. However, most existing prompt-tuning-based methods focus on designing prompts for node features while ignoring the graph structures that contain task-related information in the heterogeneous graph (Ma et al., 2024; Yu et al., 2024a). As a result, the parameters in existing prompt-tuning-based methods may be insufficient to model the input data effectively, leading to the increased training error $\hat{\mathcal{E}}_M$. Therefore, although we cannot directly find the optimal parameters $\overline{\mathcal{P}}_M$, we can better fit the input data with few parameters to decrease the training error, thus decreasing the upper bound of the test error and approaching the optimal parameters. To do this, in this paper, we propose to design dual structure-aware adapters to model both node features as well as homogeneous and heterogeneous graph structures, thereby making the parameters $\mathcal{P}_M$ closer to the optimal parameters $\overline{\mathcal{P}}_M$ to improve the generalization ability.

To do this, for the homogeneous branch in pre-trained HGNNs, we design a homogeneous adapter to capture the task-related homogeneous structural information (*i.e.,* the connections among nodes within the same type). Specifically, the homogeneous adapter includes two steps (*i.e.,* mapping and message-passing), which tune node features and the homogeneous graph structure, respectively. In the mapping step, we first employ the MLP $f_\delta$ to obtain the mapped representations $\mathbf{F}$ for the frozen representations $\tilde{\mathbf{H}}$ before message-passing, *i.e.,*

$$\mathbf{F} = \text{ReLU}(\tilde{\mathbf{H}}\mathbf{W}_\delta), \tag{4}$$

where $\mathbf{W}_\delta \in \mathbb{R}^{d \times d'}$ indicates the trainable parameters of $f_\delta$, and $\text{ReLU}(\cdot)$ indicates the ReLU activation function. We further follow the lightweight principle (Hu et al., 2021; Chen et al., 2022) to decrease the number of parameters by decomposing $\mathbf{W}_\delta$ into two low-rank matrices, *i.e.,* $\mathbf{W}_\delta = \mathbf{W}_{down}\mathbf{W}_{up}$, where $\mathbf{W}_{down} \in \mathbb{R}^{d \times t}$, $\mathbf{W}_{up} \in \mathbb{R}^{t \times d'}$, and $t \ll d, d'$.

In the message-passing step, we focus on tuning the homogeneous graph structure to capture the task-related homogeneous structural information, *i.e.,* assign large weights for node pairs within the same class while assigning small weights for node pairs from different classes. However, the homogeneous graph structure is typically not provided in the heterogeneous graph, and previous methods generally employ meta-paths to build it, which requires expert knowledge and incurs large computation costs (Jing et al., 2021; Wang et al., 2023). To alleviate this issue, we propose to tune the homogeneous graph structure adaptively by calculating the similarity weight $\tilde{\mathbf{a}}_{i,j}$ between representations of nodes $v_i$ and $v_j$ from the same node type, *i.e.,*

$$\tilde{\mathbf{a}}_{i,j} = \frac{(\tilde{\mathbf{h}}_i \mathbf{W}_\vartheta) \cdot (\tilde{\mathbf{h}}_j \mathbf{W}_\vartheta)^T}{\|\tilde{\mathbf{h}}_i \mathbf{W}_\vartheta\| \|\tilde{\mathbf{h}}_j \mathbf{W}_\vartheta\|}, \tag{5}$$

where $\tilde{\mathbf{h}}_i$ and $\tilde{\mathbf{h}}_j$ indicate the frozen representations of nodes $v_i$ and $v_j$ before message-passing, and $\mathbf{W}_\vartheta \in \mathbb{R}^{d \times d'}$ indicates the trainable parameters that also can be decomposed into low-rank matrices. To improve the performance and reduce computation costs, we only calculate $\tilde{\mathbf{a}}_{i,j}$ between node $v_i$ and its $k$ nearest neighbors to obtain a sparse graph structure $\tilde{\mathbf{A}}$. After that, we can further derive the positive and symmetric graph structure $\mathbf{A} \in \mathbb{R}^{n \times n}$ by

$$\mathbf{A} = \frac{\text{ReLU}(\tilde{\mathbf{A}}) + \text{ReLU}(\tilde{\mathbf{A}})^T}{2}, \tag{6}$$

where $n$ indicates the number of target nodes from the same node type.

In Eq. (6), a larger value of the element $\mathbf{a}_{i,j}$ indicates a stronger correlation between nodes $v_i$ and $v_j$, suggesting that they are more likely to belong to the same class. Based on the mapped representations $\mathbf{F}$ and the adaptively learned homogeneous graph structure $\mathbf{A}$, we conduct the message-passing among nodes that are likely to belong to the same class and obtain the adapted representations $\tilde{\mathbf{F}}$, *i.e.,* $\tilde{\mathbf{F}} = \mathbf{AF}$. As a result, the adapted representations $\tilde{\mathbf{F}}$ are expected to capture the adaptive homogeneous structural information and aggregate information from nodes within the same class. Subsequently, we can obtain the homogeneous representations $\tilde{\mathbf{Z}}$ by summing the adapted representations $\tilde{\mathbf{F}}$ from the homogeneous adapter and the frozen representations $\tilde{\mathbf{E}}$ after message-passing, *i.e.,*

$$\tilde{\mathbf{Z}} = \tilde{\mathbf{E}} + \alpha \tilde{\mathbf{F}}, \tag{7}$$

where $\alpha$ is non-negative parameter.

In addition to the homogeneous adapter, we further propose to design a heterogeneous adapter for the heterogeneous branch to capture the task-related heterogeneous structural information (*i.e.,* the connections among nodes from different types). Similarly, the heterogeneous adapter also consists of mapping and message-passing steps. Specifically, in the mapping step, we first employ the MLP $f_\theta$ to obtain the mapped representations $\mathbf{M}$ for the frozen representations $\hat{\mathbf{H}}$ before message-passing, *i.e.,*

$$\mathbf{M} = \text{ReLU}(\hat{\mathbf{H}}\Theta_{down}\Theta_{up}), \tag{8}$$

where $\Theta_{down} \in \mathbb{R}^{d \times t'}$ and $\Theta_{up} \in \mathbb{R}^{t' \times d'}$ indicate the trainable parameters of $f_\theta$, and $t' \ll d, d'$.

In the message-passing step, we focus on tuning the heterogeneous graph structure to capture the task-related structural information, *i.e.,* assign large weights for important neighbors while assigning small weights for unimportant neighbors. To do this, we propose to calculate the weight $\mathbf{s}_{i,r}$ for the $r$-th type of neighbor of each target node $v_i$ with the score function, thereby obtaining the heterogeneous graph structure $\mathbf{S} \in \mathbb{R}^{n \times |\mathcal{R}|}$, *i.e.,*

$$\mathbf{s}_{i,r} = \frac{\exp(\text{Tanh}(\hat{\mathbf{h}}_{i,r}\mathbf{W}_\epsilon))}{\sum_{r'=1}^{|\mathcal{R}|} \exp(\text{Tanh}(\hat{\mathbf{h}}_{i,r'}\mathbf{W}_\epsilon))}, \tag{9}$$

where $\hat{\mathbf{h}}_{i,r}$ indicates the representation of the $r$-th type of neighbor for node $v_i$, $\mathbf{W}_\epsilon \in \mathbb{R}^{d \times 1}$ indicates the trainable parameters of the score function, $\text{Tanh}(\cdot)$ denotes the Tanh activation function, and $|\mathcal{R}|$ denotes the number of edges types.

In Eq. (9), a larger value of the element $\mathbf{s}_{i,r}$ indicates that the $r$-th type of neighbor is more important for node $v_i$. With the mapped representations $\mathbf{M}$ and the adaptively learned heterogeneous graph structure $\mathbf{S}$, we conduct the message-passing among nodes from different types and obtain the adapted representations $\hat{\mathbf{M}}$, *i.e.,* $\hat{\mathbf{m}}_i = \sum_{r=1}^{|\mathcal{R}|} \mathbf{s}_{i,r}\mathbf{m}_{i,r}$, where $\mathbf{m}_{i,r}$ indicates the mapped representation of the $r$-th type of neighbor for node $v_i$. As a result, the adapted representations $\hat{\mathbf{M}}$ are expected to capture the heterogeneous structural information and aggregate information from important heterogeneous neighbors with large weights. Subsequently, we can obtain the heterogeneous representations $\hat{\mathbf{Z}}$ by summing the adapted representations $\hat{\mathbf{M}}$ from the heterogeneous adapter and the frozen representations $\hat{\mathbf{E}}$ after message-passing, *i.e.,*

$$\hat{\mathbf{Z}} = \hat{\mathbf{E}} + \beta \hat{\mathbf{M}}, \tag{10}$$

where $\beta$ is a non-negative parameter.

As a result, the dual structure-aware adapters tune node features as well as homogeneous and heterogeneous graph structures simultaneously, thereby capturing more task-related structural information

than existing prompt-tuning-based methods. We then concatenate homogeneous representations $\tilde{\mathbf{Z}}$ and heterogeneous representations $\hat{\mathbf{Z}}$ to obtain the final node representations $\mathbf{Z}$. Consequently, the final node representations $\mathbf{Z}$ aggregate information from nodes within the same class as well as important neighbors from other node types. This enables the model to better fit the input data and get closer to the optimal parameters $\overline{\mathcal{P}}_M$, thus reducing the training error and improving the model's generalization ability to solve the challenge (ii) (verified in Appendix E.3).

## 2.4 POTENTIAL LABELED DATA EXTENSION

Designing the dual structure-aware adapters encourages the parameters $\mathcal{P}_M$ to approach the optimal parameters $\overline{\mathcal{P}}_M$, thereby decreasing the training error $\hat{\mathcal{E}}_M$ and approaching the lowest generalization error bound. Actually, based on Theorem 2.3, there is another way to further decrease the generalization error bound, *i.e.,* increasing the number of labeled data (*i.e.,* $n_M$) in training set to decrease the generalization gap bound $O(\sqrt{|\overline{\mathcal{P}}_M|/n_M})$. However, obtaining a large number of labeled data is challenging and costly in real scenarios. To solve this issue, in this paper, we design a label-propagated contrastive loss and two self-supervised losses, extending all unlabeled nodes as the potential labeled data to further improve the model's generalization ability.

Specifically, we first propose to conduct the label propagation for the unlabeled data based on the adaptively tuned homogeneous graph structure $\mathbf{A}$ and the label matrix $\mathbf{Y}$, *i.e.,*

$$\tilde{\mathbf{Y}} = \mathbf{A}\mathbf{Y}, \tag{11}$$

where $\mathbf{Y}$ consists of one-hot label indicator vectors for labeled nodes and zero vectors for unlabeled nodes. As a result, Eq. (11) encourages the label propagation from the labeled nodes and tends to assign the same label for unlabeled nodes within the same class. Note that we utilize the homogeneous graph structure for the label propagation instead of the heterogeneous graph structure, as the label propagation occurs only among target nodes of the same type.

Given the propagated labels $\tilde{\mathbf{Y}}$, a common approach is to train a classifier that maps node representations $\mathbf{Z}$ to a probability matrix and then computes the cross-entropy loss between the label matrix and the probability matrix. However, the cross-entropy loss typically differs from the objective function used in the pre-training stage, leading to the gap between pre-trained models and downstream tasks (Jing et al., 2021; Yu et al., 2024c). Fortunately, existing literature indicates that any standard contrastive pre-training task on graphs can be reformulated as the objective function based on subgraph similarity (Liu et al., 2023b; Yu et al., 2024b). Therefore, we propose to bridge the gap between different pre-trained models and downstream tasks by reformulating the downstream objective function according to the class-subgraph similarity.

To do this, we first employ a projection $g_\rho$ to map node representations $\mathbf{Z}$, resulting in the prediction matrix $\mathbf{P}$ of all nodes, *i.e.,* $\mathbf{P} = \mathbf{Z}\mathbf{W}_\rho$, where $\mathbf{W}_\rho \in \mathbb{R}^{d' \times c}$ is the trainable parameters of $g_\rho$, and $c$ denotes the number of classes. We then consider nodes with the same propagated label to be in the same subgraph, and we can further obtain the class-subgraph prediction $\mathbf{c}_{\tilde{\mathbf{y}}_i}$ by averaging the prediction vectors of nodes whose original labels equal to $\tilde{\mathbf{y}}_i$. After that, we propose a contrastive loss based on the subgraph similarity to incorporate supervision signals by enforcing the node prediction close to its class-subgraph prediction while far away from different class-subgraph predictions, *i.e.,*

$$\mathcal{L}_{con} = -\sum_i^{\tilde{\mathbf{Y}}_L} \ln \frac{\exp(\mathrm{sim}(\mathbf{p}_i, \mathbf{c}_{\tilde{\mathbf{y}}_i})/\tau)}{\sum_{\tilde{\mathbf{y}}_i \neq \tilde{\mathbf{y}}_j} \exp(\mathrm{sim}(\mathbf{p}_i, \mathbf{c}_{\tilde{\mathbf{y}}_j})/\tau)} - \lambda \sum_i^{\tilde{\mathbf{Y}}_{UL}} \ln \frac{\exp(\mathrm{sim}(\mathbf{p}_i, \mathbf{c}_{\tilde{\mathbf{y}}_i})/\tau)}{\sum_{\tilde{\mathbf{y}}_i \neq \tilde{\mathbf{y}}_j} \exp(\mathrm{sim}(\mathbf{p}_i, \mathbf{c}_{\tilde{\mathbf{y}}_j})/\tau)}, \tag{12}$$

where $\lambda$ is a non-negative parameter, $\tilde{\mathbf{Y}}_L$ and $\tilde{\mathbf{Y}}_{UL}$ indicate the sets of node indices with and without original labels, respectively, $\mathrm{sim}(\cdot)$ denotes a similarity function between two vectors, and $\tau$ is a temperature parameter. Consequently, Eq. (12) simulates objective functions in different pre-trained HGNN models based on the subgraph similarity, thereby bridging the gap between pre-trained models and downstream tasks. This facilitates the application of the proposed method to different pre-trained HGNN models. Moreover, Eq. (12) encourages the prediction vectors of both labeled and unlabeled nodes to effectively capture the class information, potentially increasing the number of labeled data to further decrease the model's generalization error bound.

Although the label-propagated contrastive loss increases the number of labeled data potentially, the confidence of propagated labels of unlabeled nodes in $\tilde{\mathbf{Y}}$ may be insufficient during the early stages

of training due to the gradual optimization of the graph structure $\mathbf{A}$. This may hinder the optimization of the dual adapters, *i.e.,* the homogeneous and heterogeneous graph structures of labeled nodes may be better tuned than that of unlabeled nodes. To address this issue, we start with the essence of homogeneous and heterogeneous graph structures, and then design two self-supervised losses for the dual adapters and treat both labeled and unlabeled nodes as equal supervision signals.

Intuitively, for nodes within the same type, a good homogeneous graph structure $\mathbf{A}$ should connect nodes from the same class while disconnecting nodes from different classes. Moreover, existing literature suggests that nodes with similar node features generally come from the same class (Liu et al., 2022; Wu et al., 2023). For instance, in an academic heterogeneous graph, if the features of two "paper" nodes share many keywords, they are likely to belong to the same class. Therefore, there is an intuitive way to optimize the homogeneous graph structure by connecting nodes that share similar node features while disconnecting nodes with dissimilar node features. To do this, we design a feature reconstruction loss to align node features before and after the message-passing, thus optimizing the graph structure $\mathbf{A}$ to assign appropriate weights to node pairs within the same class, *i.e.,*

$$\mathcal{L}_{rec} = -\sum_{i=1}^{m}\sum_{j=1}^{f} \mathbf{x}_{i,j} \ln(\mathbf{A}\mathbf{X})_{i,j}, \tag{13}$$

where $m$ and $f$ denote the number of sampled nodes and the feature dimension, respectively. Eq. (13) enforces the reconstructed node features after message-passing to be aligned with the original node features, which requires that message-passing occurs only among nodes with similar node features. Therefore, Eq. (13) encourages the graph structure $\mathbf{A}$ to connect nodes within the same class while disconnecting nodes from different classes as much as possible to optimize it.

Different from tuning the homogeneous graph structure that connects nodes within the same type, the heterogeneous graph structure connects nodes from different types. Therefore, the feature reconstruction may not be suitable for optimizing the heterogeneous graph structure $\mathbf{S}$ as the connected nodes come from different feature distributions. To alleviate this, we first analyze the significance of different heterogeneous neighbors and then optimize the weights assigned to them.

Intuitively, if a certain type of heterogeneous neighbor of node $v_i$ provides more relevant information for identifying the node's label than other neighbors, then such neighbors are more important to the node $v_i$ and should be assigned larger weights. That is, although heterogeneous neighbors come from different feature distributions, their representations may still partially overlap with the class-subgraph representation $\mathbf{c}_{\tilde{\mathbf{y}}_i}$ of the node $v_i$ to provide class-related information. Furthermore, the degree of overlap is greater for important neighbors than unimportant neighbors. Therefore, this paper designs a margin loss to relatively narrow the distance between the class-subgraph representation $\mathbf{c}_{\tilde{\mathbf{y}}_i}$ and the adapted representation $\hat{\mathbf{m}}_i$, thus optimizing the graph structure $\mathbf{S}$ to assign large weights for important heterogeneous neighbors, *i.e.,*

$$\mathcal{L}_{mar} = \sum_{i,j=1,\tilde{\mathbf{y}}_i \neq \tilde{\mathbf{y}}_j}^{n} \left\{ d(\mathbf{c}_{\tilde{\mathbf{y}}_i}, \hat{\mathbf{m}}_i)^2 - d(\mathbf{c}_{\tilde{\mathbf{y}}_i}, \hat{\mathbf{m}}_j)^2 + \gamma \right\}_+, \tag{14}$$

where $\{\cdot\}_+ = \max\{\cdot, 0\}$, $d(\cdot)$ indicates a distance measurement between two vectors, and $\gamma$ is a non-negative parameter. Eq. (14) aims to decrease the distance $d(\mathbf{c}_{\tilde{\mathbf{y}}_i}, \hat{\mathbf{m}}_i)^2$ while increasing the distance $d(\mathbf{c}_{\tilde{\mathbf{y}}_i}, \hat{\mathbf{m}}_j)^2$, thus ensuring the "safe" margin $\gamma$ between them. Therefore, the adapted representation $\hat{\mathbf{m}}_i = \sum_{r=1}^{|\mathcal{R}|} \mathbf{s}_{i,r} \mathbf{m}_{i,r}$ is encouraged to preserve its class-related information that shared with $\mathbf{c}_{\tilde{\mathbf{y}}_i}$. This optimizes the heterogeneous graph structure $\mathbf{S}$ to assign large weights to important neighbors that contain more class-related information. Note that the margin loss only requires the relative distance to exceed $\gamma$, unlike traditional contrastive losses (*e.g.,* InfoNCE (Oord et al., 2018)) that enforce the alignment between the class-subgraph representation and the adapted representation. Such direct alignment may be inappropriate, as these representations come from different feature distributions (verified in Appendix E.8).

Finally, we integrate the contrastive loss in Eq. (12), the feature reconstruction loss in Eq. (13), and the margin loss in Eq. (14) to obtain the objective function of the proposed method, *i.e.,*

$$\mathcal{J} = \mathcal{L}_{con} + \eta\mathcal{L}_{rec} + \mu\mathcal{L}_{mar}, \tag{15}$$

where $\eta$ and $\mu$ are non-negative parameters. With the objective function in Eq. (15), the proposed method optimizes the dual adapters and extends the potential labeled data by incorporating both

Table 1: Classification performance (*i.e.,* Macro-F1 and Micro-F1) on all heterogeneous graph datasets, where the best results are highlighted in bold, while improved results with the proposed HG-Adapter are underlined. The "+" symbol indicates the integration of HG-Adapter and HetGPT with original pre-trained HGNN models.

| Method | ACM | | Yelp | | DBLP | | Aminer | |
|---|---|---|---|---|---|---|---|---|
| | Macro-F1 | Micro-F1 | Macro-F1 | Micro-F1 | Macro-F1 | Micro-F1 | Macro-F1 | Micro-F1 |
| HAN | 89.4±0.2 | 89.2±0.2 | 90.5±1.2 | 90.7±1.4 | 91.2±0.4 | 92.0±0.5 | 65.3±0.7 | 72.8±0.4 |
| HGT | 91.5±0.7 | 91.6±0.6 | 89.9±0.5 | 90.2±0.6 | 90.9±0.6 | 91.7±0.8 | 64.5±0.5 | 71.0±0.7 |
| DMGI | 89.8±0.1 | 89.8±0.1 | 82.9±0.8 | 85.8±0.9 | 92.1±0.2 | 92.9±0.3 | 63.8±0.4 | 67.6±0.5 |
| HGCML | 90.6±0.7 | 90.7±0.5 | 90.7±0.8 | 91.0±0.7 | 91.9±0.8 | 93.2±0.7 | 70.5±0.4 | 76.3±0.6 |
| HGMAE | 90.5±0.5 | 90.6±0.7 | 90.5±0.7 | 90.7±0.5 | 92.9±0.5 | 93.4±0.6 | 72.3±0.9 | 80.3±1.2 |
| HGPrompt | 92.1±0.7 | 92.0±0.6 | 92.5±0.4 | 92.3±0.5 | 93.5±0.6 | 94.0±0.7 | 74.8±0.8 | 83.9±0.6 |
| HDMI | 90.1±0.3 | 90.1±0.3 | 80.7±0.6 | 84.0±0.9 | 91.3±0.2 | 92.2±0.5 | 65.9±0.4 | 71.7±0.6 |
| +HetGPT | 91.0±0.7 | 90.9±0.6 | 81.4±0.4 | 84.8±0.5 | 91.9±0.8 | 92.9±0.6 | 67.1±0.3 | 72.9±0.4 |
| +HG-Adpater | 91.4±0.6 | 91.5±0.7 | 82.0±0.8 | 85.5±1.1 | 92.4±0.5 | 93.3±0.7 | 67.9±0.3 | 73.6±0.4 |
| HeCo | 88.3±0.3 | 88.2±0.2 | 85.3±0.7 | 87.9±0.6 | 91.0±0.3 | 91.6±0.2 | 71.8±0.9 | 78.6±0.7 |
| +HetGPT | 88.5±0.4 | 88.4±0.6 | 85.9±0.8 | 88.6±0.9 | 91.5±0.6 | 92.2±0.5 | 72.1±0.7 | 79.0±0.4 |
| +HG-Adpater | 89.0±0.5 | 89.0±0.4 | 86.4±0.6 | 89.2±0.7 | 92.3±0.8 | 92.6±0.6 | 72.8±0.5 | 79.8±0.3 |
| HERO | 92.2±0.5 | 92.1±0.7 | 92.4±0.7 | 92.3±0.6 | 93.8±0.6 | 94.4±0.4 | 75.1±0.7 | 84.5±0.9 |
| +HetGPT | 92.4±0.4 | 92.2±0.3 | 92.6±0.5 | 92.4±0.7 | 93.8±0.4 | 94.5±0.3 | 75.7±0.6 | 85.2±0.8 |
| +HG-Adapter | **92.7±0.4** | **92.7±0.7** | **93.1±0.6** | **92.7±0.5** | **94.0±0.7** | **94.7±0.8** | **78.3±0.5** | **87.1±0.6** |

labeled and unlabeled nodes as supervision signals. As a result, this decreases the generalization gap and thus further decreases the generalization error bound of existing methods to solve the challenge (iii) (verified in Appendix E.4). Actually, we can derive that the proposed method with dual adapters and labeled data extension achieves a lower generalization error bound than existing methods by decreasing the training error and the generalization gap, which can be found in Appendix C.2. Consequently, the proposed method is expected to obtain better generalization and effectiveness on downstream tasks. (verified in Section 3.2.1 and Section 3.2.2).

## 3 EXPERIMENTS

In this section, we conduct experiments on four public heterogeneous graph datasets to evaluate the proposed HG-Adapter in terms of different downstream tasks (*i.e.,* node classification and node clustering), compared to both fine-tuning-based methods and prompt-tuning-based methods. The code of the proposed method is released at `https://github.com/YujieMo/HG-Adapter`.

### 3.1 EXPERIMENTAL SETUP

#### 3.1.1 DATASETS

The used datasets include three academic datasets (*i.e.,* ACM (Wang et al., 2019), DBLP (Wang et al., 2019), and Aminer (Hu et al., 2019)), and one business dataset (*i.e.,* Yelp (Lu et al., 2019)).

#### 3.1.2 COMPARISON METHODS

The comparison methods include two traditional semi-supervised methods (*i.e.,* HAN (Wang et al., 2019) and HGT (Hu et al., 2020b)), six fine-tuning-based methods (*i.e.,* DMGI (Park et al., 2020), HDMI (Jing et al., 2021), HeCo (Wang et al., 2021), HGCML (Wang et al., 2023), HGMAE (Tian et al., 2023), and HERO (Mo et al., 2024)), and two prompt-tuning-based methods (*i.e.,* HGPrompt (Yu et al., 2024a) and HetGPT (Ma et al., 2024)), where the pre-training and the prompt-tuning of HGPrompt are both specifically designed, while HetGPT only designs the prompt-tuning and thus can be used for different pre-trained HGNN models.

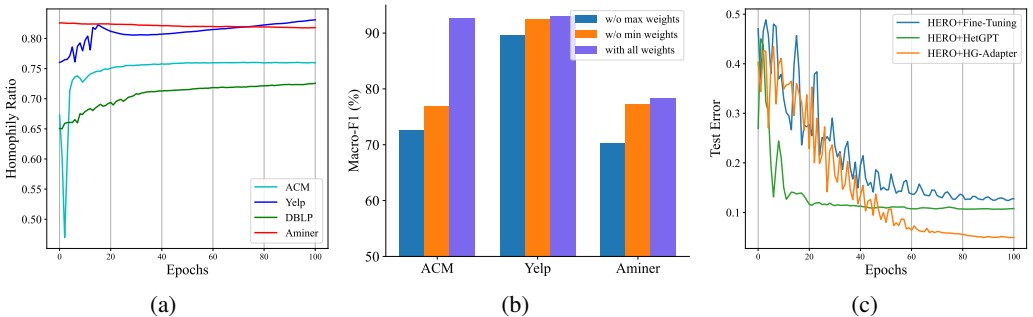

Figure 2: (a) Homophily ratios of the homogeneous graph structure $\mathbf{A}$ learned by HERO+HG-Adapter on four datasets. (b) Node classification results without maximal/minimal weights neighbors in the heterogeneous graph structure $\mathbf{S}$ learned by HERO+HG-Adapter on three datasets (excluding the DBLP dataset, as its target node has only one type of neighbors). (c) Test errors of HERO with different tuning methods (*i.e.,* traditional fine-tuning, prompt-tuning-based HetGPT, and the proposed HG-Adapter) on the ACM dataset.

## 3.2 RESULTS ANALYSIS

### 3.2.1 EFFECTIVENESS ON DOWNSTREAM TASKS

To evaluate the effectiveness of the proposed HG-Adapter, we follow previous works (Fu et al., 2020; Ma et al., 2024; Mo et al., 2024) to employ node classification and node clustering as downstream tasks and report their results in Table 1 and Appendix E, respectively. Obviously, the proposed method consistently demonstrates superior performance on both node classification task and node clustering task than other comparison methods.

Specifically, first, for the node classification task, the proposed method always outperforms fine-tuning-based and prompt-tuning-based comparison methods by large margins. For instance, the proposed method on average, improves by 0.81%, compared to the best prompt-tuning-based method (*i.e.,* HetGPT), on different pre-trained HGNN models (*i.e.,* HDMI, HeCo, and HERO). This improvement can be attributed to the dual structure-aware adapters and the potential labeled data extension, which decrease the training error and the generalization gap. Consequently, the proposed method enhances the model's generalization ability on downstream tasks to decrease the generalization error on test data, thus improving the performance of different pre-trained HGNN models.

Second, for the node clustering task, the proposed method also obtains promising improvements. For instance, the proposed method on average, improves by 2.69%, compared to the best prompt-tuning-based method (*i.e.,* HetGPT), across different pre-trained HGNN models. This demonstrates the superiority of the proposed method, which designs a contrastive loss based on the class-subgraph similarity to ensure that nodes within the same class are close to each other, thereby enhancing the clustering performance. As a result, the effectiveness of the proposed method is verified on both node classification and node clustering downstream tasks.

### 3.2.2 EFFECTIVENESS OF DUAL-ADAPTERS

To verify the effectiveness of the proposed dual structure-aware adapters, we calculate the homophily ratio (*i.e.,* the ratio of edges that connect nodes within the same class) of $\mathbf{A}$, collect the performance degradation without minimal and maximal weights neighbors in $\mathbf{S}$, evaluate the test errors of HERO with different tuning methods, and report the results in Figure 2.

First, from Figure 2(a), the homophily ratio of the learned homogeneous graph structure $\mathbf{A}$ increases rapidly at the beginning of training and tends to stabilize after reaching a high value. This suggests that the homogeneous adapter effectively learns $\mathbf{A}$ to build edges among nodes within the same class. As a result, the homogeneous adapter captures the task-related homogeneous structural information and promotes message-passing within the same class. Second, from Figure 2(b), the proposed HG-Adapter without minimal weights neighbors obtains significantly less performance degradation than without

Table 2: Classification performance (*i.e.,* Macro-F1 and Micro-F1) of each component in the objective function $\mathcal{J}$ on all heterogeneous graph datasets.

| $\mathcal{L}_{con}$ | $\mathcal{L}_{rec}$ | $\mathcal{L}_{mar}$ | ACM | | Yelp | | DBLP | | Aminer | |
|---|---|---|---|---|---|---|---|---|---|---|
| | | | Macro-F1 | Micro-F1 | Macro-F1 | Micro-F1 | Macro-F1 | Micro-F1 | Macro-F1 | Micro-F1 |
| − | − | ✓ | 30.2±1.1 | 39.5±1.3 | 50.3±0.7 | 56.7±0.9 | 11.9±0.5 | 31.4±0.7 | 35.7±0.8 | 67.7±1.0 |
| − | ✓ | − | 32.3±0.9 | 40.5±0.8 | 41.7±0.6 | 52.2±0.5 | 10.9±0.8 | 30.8±0.9 | 35.6±0.7 | 67.5±0.6 |
| ✓ | − | − | 87.9±0.5 | 87.8±0.7 | 89.1±0.4 | 89.0±0.3 | 80.7±0.8 | 82.6±0.7 | 73.2±0.5 | 79.8±0.3 |
| − | ✓ | ✓ | 32.3±0.5 | 40.5±0.6 | 40.6±0.8 | 48.2±0.7 | 11.9±0.5 | 31.4±0.4 | 35.7±0.6 | 67.7±0.7 |
| ✓ | − | ✓ | 89.6±0.5 | 89.5±0.7 | 90.0±0.6 | 89.6±0.4 | 91.6±0.9 | 92.5±1.1 | 75.1±0.6 | 82.2±0.7 |
| ✓ | ✓ | − | 90.1±0.4 | 90.0±0.5 | 92.0±0.5 | 91.8±0.6 | 92.7±0.8 | 93.5±0.7 | 76.3±0.5 | 84.6±0.4 |
| ✓ | ✓ | ✓ | **92.7±0.4** | **92.7±0.7** | **93.1±0.6** | **92.7±0.5** | **94.0±0.7** | **94.7±0.8** | **78.3±0.5** | **87.1±0.6** |

maximal weights neighbors. This demonstrates that the heterogeneous adapter effectively learns the heterogeneous graph structure $\mathbf{S}$ to assign large weights for important neighbors while assigning small weights for unimportant neighbors. Therefore, the heterogeneous adapter enables nodes to aggregate information from important neighbors, thus capturing more class-related information to benefit downstream tasks. Third, from Figure 2(c), the proposed HG-Adapter achieves the lowest test error than fine-tuning-based and prompt-tuning-based methods on the test data. This indicates that the proposed method indeed achieves lower generalization errors than existing methods on unlabeled data, obtaining better generalization ability on downstream tasks.

### 3.2.3 ABLATION STUDY

The proposed method investigates the objective function $\mathcal{J}$ to optimize the dual adapters and extend the labeled data potentially. To verify the effectiveness of each component of $\mathcal{J}$ (*i.e.,* the contrastive loss $\mathcal{L}_{con}$, the feature reconstruction loss $\mathcal{L}_{rec}$, and the margin loss $\mathcal{L}_{mar}$), we investigate the performance of all variants on the node classification task and report the results in Table 2.

From Table 2, we have the observations as follows. First, the variant without $\mathcal{L}_{con}$ performs significantly worse to the other two variants (*i.e.,* without $\mathcal{L}_{rec}$ and without $\mathcal{L}_{mar}$, respectively). The reason can be attributed to the fact that the label information is necessary for the proposed HG-Adapter because it provides the ground truth for optimizing adapters and node predictions. Second, the proposed method with the complete objective function obtains the best performance. For example, the proposed method on average improves by 2.0%, compared to the best variant (*i.e.,* without $\mathcal{L}_{mar}$), indicating that all components in the objective function are necessary for the proposed method. This is reasonable because the feature reconstruction loss $\mathcal{L}_{rec}$ and the margin loss $\mathcal{L}_{mar}$ optimize graph structures and incorporate all unlabeled nodes as equal supervision signals, thus improving the model generalization. These observations are consistent with our claims, *i.e.,* capturing the task-related structural information with dual adapters and extending the potential labeled data is essential for improving the effectiveness and generalization ability of pre-trained HGNN models.

## 4 CONCLUSION

In this paper, we derived the generalization error bound for existing prompt-tuning-based methods and then proposed a novel framework with dual structure-aware adapters and potential labeled data extension to address existing issues. Specifically, we first established the theoretical foundation by deriving the generalization error bound for existing methods based on the training error and the generalization gap. We then designed homogeneous and heterogeneous adapters to adaptively tune homogeneous and heterogeneous graph structures, thus capturing the task-related structural information to decrease the training error. Moreover, we designed a label-propagated contrastive loss and two self-supervised losses on unlabeled nodes, thus optimizing dual adapters and potentially increasing the labeled data to decrease the generalization gap. Theoretical analysis indicates that the proposed method is expected to obtain a lower generalization error bound and better generalization ability than existing prompt-tuning-based methods. Comprehensive experiments verify the effectiveness and generalization of the proposed method on various heterogeneous graph datasets and different downstream tasks. We discuss potential limitations and future directions in Appendix F.

ACKNOWLEDGMENTS

This project is supported by the National Key Research and Development Program of China under Grant No. 2022YFA1004100, the Natural Science Foundation of Guangdong Province of China under Grant No. 2024A1515011381, and National Research Foundation, Singapore, under its AI Singapore Programme (AISG Award No: AISG2-RP-2021-023).

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

# A  RELATED WORK

This section briefly reviews topics related to this work, including fine-tuning-based pre-trained heterogeneous graph neural networks in Section A.1, and graph prompt-tuning and adapter-tuning in Section A.2.

## A.1  FINE-TUNING-BASED PRE-TRAINED HETEROGENEOUS GRAPH NEURAL NETWORKS

In recent years, inspired by the prosperity of pre-trained models in the fields of computer vision (Bao et al., 2021) and natural language processing (Dong et al., 2019), many pre-trained heterogeneous graph neural networks (HGNNs) have been introduced for the heterogeneous graph data (Wang et al., 2019; Jiang et al., 2021). Generally, these pre-trained HGNN models are trained in a self-supervised fashion, facilitating the transfer of knowledge to downstream tasks through a fine-tuning step (Jing et al., 2021; Zhu et al., 2022).

Existing fine-tuning-based pre-trained HGNN methods can be broadly classified into two groups, *i.e.,* meta-path-based methods and meta-path-free methods. In meta-path-based methods, several graphs are usually constructed based on different pre-defined meta-paths to examine diverse relationships among nodes that share similar labels (Jing et al., 2021; Zhu et al., 2022). For example, STENCIL (Zhu et al., 2022) and HDMI (Jing et al., 2021) construct meth-path-based graphs and then conduct node-level consistency constraints (*e.g.,* contrastive loss) between node representations

in different graphs. In addition, HGCML (Wang et al., 2023) and CPIM (Mo et al., 2023) propose to maximize the mutual information between node representations from different meta-path-based graphs. However, pre-defined meta-paths in these methods generally require expert knowledge and prohibitive computation costs (Zhang et al., 2022). Therefore, meta-path-free methods are proposed to capture the relationships among nodes without meta-paths. For example, SR-RSC (Zhang et al., 2022) designs a multi-hop contrast to optimize the regional structural information by utilizing the strong correlation between nodes and their neighbor graphs. In addition, recently, HERO (Mo et al., 2024) made the first attempt to learn an adaptive self-expressive matrix to capture the homophily in the heterogeneous graph, thus avoiding meta-paths.

### A.2 Graph Prompt-tuning and Adapter-tuning

The "pre-train, fine-tuning" paradigm has become prevalent in graph learning, particularly for tasks with limited labeled data (Ma et al., 2024). However, this approach often suffers from a mismatch between the objectives of pre-training and downstream tasks, resulting in a "negative transfer" problem. Consequently, the knowledge acquired during the pre-training stage can negatively impact the performance of downstream tasks.

To solve this issue, recent research suggests the "pre-train, prompt-tuning" diagram to establish a connection between downstream tasks and pre-trained models by designing a learnable prompt that modifies the model input (Fang et al., 2023; Yu et al., 2024b). For example, for the homogeneous graph, GPPT (Sun et al., 2022) introduces prompt templates to align the link prediction pre-training task with the downstream node classification task. ProG (Sun et al., 2023) proposes a new multi-task prompting method for graph models by unifying the format of graph prompts and language prompts with the prompt token, token structure, and inserting pattern. GraphPrompt (Liu et al., 2023b) employs subgraph similarity as its template and designs a learnable prompt to unify pre-training with multiple downstream tasks. For the heterogeneous graph, HGPrompt (Yu et al., 2024a) proposes to unify pre-training and downstream tasks as well as homogeneous and heterogeneous graphs via dual-template and dual-prompt design. HetGPT (Ma et al., 2024) designs virtual class and heterogeneous feature prompts, and reformulates downstream tasks to mirror pretext tasks. Despite effectiveness, both HGPrompt and HetGPT ignore the graph structures during the prompt-tuning stage. Although part homogeneous graph prompt-tuning methods (*e.g.,* ProG) investigate to design structure prompts, they may not easily be transferred to the heterogeneous graph, as the more complex graph structures in the heterogeneous graph. As a result, when tuning the pre-trained HGNNs, the model may not sufficiently model the input data to increase the training error and decrease the generalization ability.

Different from the "pre-train, prompt-tuning" diagram, the "pre-train, adapter-tuning" diagram aims to insert adapter modules with lightweight neural network architecture to bridge the gap between pre-trained models and downstream tasks. Recent works propose adapter-tuning for homogeneous graph pre-trained models by inserting different adapter modules (Li et al., 2024; Gui et al., 2024). For example, G-Adapter (Gui et al., 2024) leverages the graph structure and Bregman proximal point optimization strategy to mitigate the feature distribution shift issue. In addition, AdapterGNN (Li et al., 2024) proposes to preserve the knowledge of the large pre-trained model and leverage highly expressive adapters for graph neural networks, adapting to downstream tasks effectively with only a few parameters. However, they are designed for the homogeneous graph and cannot easily transfer to the heterogeneous graph. Moreover, these adapter-tuning-based methods also cannot tune the graph structures, thus increasing the training error and decreasing the model generalization.

As a result, when tuning pre-trained HGNN models, despite the effectiveness of existing prompt-tuning-based methods, they still have some limitations to address. That is, homogeneous and heterogeneous graph structures are generally ignored in the tuning stage, leading to increased training error and decreased generalization ability. Moreover, existing prompt-tuning-based methods may suffer from the issue of limited labeled data in the tuning stage, leading to a large generalization gap between training and test errors and sub-optimal generalization on downstream tasks.

## B Algorithm

This section provides the pseudo-code of the proposed method in Section B.1 and the complexity analysis in Section B.2.

## B.1 ALGORITHM

---

**Algorithm 1** The pseudo-code of the proposed method.

---

**Input:** Heterogeneous graph $\mathbf{G} = (\mathcal{V}, \mathcal{E}, \mathcal{A}, \mathcal{R}, \phi, \varphi)$, pre-trained HGNN model, maximum training steps $E$;

**Output:** Homogeneous and heterogeneous adapters, projection $g_\rho$;

 1: Initialize parameters and upload pre-trained parameters;
 2: **while** not reaching $E$ **do**
 3:     Obtain homogeneous graph structure $\mathbf{A}$ by Eq. (7);
 4:     Obtain the homogeneous representations $\tilde{\mathbf{Z}}$ by Eq. (7);
 5:     obtain heterogeneous graph structure $\mathbf{S}$ by Eq. (9);
 6:     Obtain the heterogeneous representations $\hat{\mathbf{Z}}$ by Eq. (10);
 7:     Conduct the label propagation by Eq. (11);
 8:     Conduct the label-propagated contrastive loss by Eq. (12);
 9:     Conduct the feature reconstruction loss by Eq. (13);
10:     Conduct the margin loss by Eq. (14);
11:     Compute the objective function $\mathcal{J}$ by Eq. (15);
12:     Back-propagate $\mathcal{J}$ to update model weights;
13: **end while**

---

## B.2 COMPLEXITY ANALYSIS

The proposed method consists of two prats, i.e., dual structure-aware adapters and potential labeled data extension. We analyze the time complexity of each part as follows. First, the time complexity of the dual structure-aware adapters is $\mathcal{O}(nkd + n|\mathcal{R}|)$, where $n$, $k$, $d$, and $|\mathcal{R}|$ indicate the number of nodes, the number of neighbors of each node, the number representation dimension, and the number of edge types, respectively. Second, the time complexity of the potential labeled data extension is $\mathcal{O}(nkc + nc^2 + nkf)$, where $c$ and $f$ indicate the number of classes and dimensions of node features, respectively. Therefore, the overall time complexity of the proposed HG-Adapter is $\mathcal{O}(n(kd + |\mathcal{R}| + kc + c^2 + kf))$. As a result, The proposed HG-Adapter is scaled linearly with the sample size and has the potential to be implemented with limited resources.

## C  PROOFS OF THEOREMS

### C.1  PROOF OF THEOREM 2.3

To prove Theorem 2.3, we first introduce the generalization error bound for finite hypothesis space in classical regime (Bousquet & Elisseeff, 2000; Huang & Meyn, 2013) by the following Lemma.

**Lemma C.1.** *(Generalization error bound for finite hypothesis space in the classical regime.) Statistically, the upper bound $\mathcal{U}(\mathcal{E})$ of the test error $\mathcal{E}$ of a model in the finite hypothesis space is determined as follows:*

$$\mathcal{E} \leq \mathcal{U}(\mathcal{E}) = \hat{\mathcal{E}}(\mathcal{D}_n, \mathcal{P}) + O(\sqrt{|\mathcal{P}|/n}), \tag{16}$$

*where training data $\mathcal{D}_n$ and parameters $\mathcal{P}$ are variables of the training error $\hat{\mathcal{E}}$. The number of training samples $n$ and the size of parameter space $|\mathcal{P}|$ are variables of the generalization gap bound $O(\sqrt{|\mathcal{P}|/n})$ between the training error and the test error.*

*Proof.* Let $\mathcal{H}$ represent a finite hypothesis space, where each $h \in \mathcal{H}$ corresponds to a set of trained parameters within this parameter space $\mathcal{H}$. $\hat{\mathcal{E}}_{\mathcal{D}_n}$ represents training error over sampled training data $\mathcal{D}_n$, and $\mathcal{E}$ represents test error. $n$ is the number of training data.

Then, the probability that the difference between the test error and the training error of the hypothesis space $h$ is greater than $\varepsilon$ can be written as

$$P(\exists h \in \mathcal{H}, |\hat{\mathcal{E}}_{\mathcal{D}_n}(h) - \mathcal{E}(h)| > \varepsilon). \tag{17}$$

Based on the union bound of probability, we further have:

$$
\begin{aligned}
&P(\exists h \in \mathcal{H}, |\hat{\mathcal{E}}_{\mathcal{D}_n}(h) - \mathcal{E}(h)| > \varepsilon) \\
&= P([|\hat{\mathcal{E}}_{\mathcal{D}_n}(h_1) - \mathcal{E}(h_1)| > \varepsilon] \vee \cdots \vee [|\hat{\mathcal{E}}_{\mathcal{D}_n}(h_{|\mathcal{H}|}) - \mathcal{E}(h_{|\mathcal{H}|})| > \varepsilon]) \\
&\leq \sum_{h \in \mathcal{H}} P(|\hat{\mathcal{E}}_{\mathcal{D}_n}(h) - \mathcal{E}(h)| > \varepsilon).
\end{aligned}
\tag{18}
$$

In addition, based on the Hoeffding's Inequality, we have

$$
\sum_{h \in \mathcal{H}} P(|\hat{\mathcal{E}}_{\mathcal{D}_n}(h) - \mathcal{E}(h)| > \varepsilon) \leq 2 \exp(-2n\varepsilon^2).
\tag{19}
$$

We then let $2 \exp(-2n\varepsilon^2) = \delta/|\mathcal{H}|$, where $0 < \delta < 1$ and have

$$
\sum_{h \in \mathcal{H}} P(|\hat{\mathcal{E}}_{\mathcal{D}_n}(h) - \mathcal{E}(h)| > \varepsilon) \leq \sum_{h \in \mathcal{H}} \delta/|\mathcal{H}| \leqslant |\mathcal{H}| \cdot \delta/|\mathcal{H}| = \delta.
\tag{20}
$$

That is, when $2 \exp(-2n\varepsilon^2) = \delta/|\mathcal{H}|$, we have

$$
\sum_{h \in \mathcal{H}} P(|\hat{\mathcal{E}}_{\mathcal{D}_n}(h) - \mathcal{E}(h)| > \varepsilon) \leq \delta.
\tag{21}
$$

According to $2 \exp(-2n\varepsilon^2) = \delta/|\mathcal{H}|$, we can obtain $\varepsilon = \sqrt{\frac{\ln|\mathcal{H}| + \ln(2/\delta)}{2n}}$. Moreover, we can rewrite Eq. (21) as

$$
P(\forall h \in \mathcal{H}, |\hat{\mathcal{E}}_{\mathcal{D}_n}(h) - \mathcal{E}(h)| \leq \varepsilon) \geq 1 - \delta.
\tag{22}
$$

Replace $\varepsilon$ with $\sqrt{\frac{\ln|\mathcal{H}| + \ln(2/\delta)}{2n}}$, we can further have

$$
P\left(\forall h \in \mathcal{H}, |\hat{\mathcal{E}}_{\mathcal{D}_n}(h) - \mathcal{E}(h)| \leq \sqrt{\frac{\ln|\mathcal{H}| + \ln(2/\delta)}{2n}}\right) \geq 1 - \delta.
\tag{23}
$$

Therefore, with probability at least $1 - \delta$, we have

$$
\mathcal{E}(h) \leq \hat{\mathcal{E}}_{\mathcal{D}_n}(h) + \sqrt{\frac{\ln|\mathcal{H}| + \ln(2/\delta)}{2n}}.
\tag{24}
$$

For simplicity, we omit the probability notation and use $\mathcal{U}$ to represent the upper bound. We also omit the $\ln$ term and use $\mathcal{P}$ to represent the parameter space. Therefore, $|\mathcal{P}|$ quantifies the size of the parameter space, sampled data $\mathcal{D}_n$ and the trained parameters $\mathcal{P}$ are variables of the training error $\hat{\mathcal{E}}$, and we have

$$
\mathcal{E} \leq \mathcal{U}(\mathcal{E}) = \hat{\mathcal{E}}(\mathcal{D}_n, \mathcal{P}) + O(\sqrt{|\mathcal{P}|/n}).
\tag{25}
$$

Thus we complete the proof. $\qquad\square$

Based on Lemma C.1, we can further derive the lowest generalization error bound for pre-trained HGNN models during the prompt-tuning stage, which benefit from the parameter optimization by the pre-training stage.

**Theorem C.2.** *(Generalization error bound for prompt-tuning-based methods.) Statistically, the upper bound $\mathcal{U}(\mathcal{E}_M)$ of the test error $\mathcal{E}_M$ of a pre-trained HGNN model with prompt-tuning can be determined as follows:*

$$
\mathcal{U}(\mathcal{E}_M) = \hat{\mathcal{E}}_M(\mathcal{D}_{n_M}, \mathcal{P}_M) + O(\sqrt{|\mathcal{P}_M|/n_M}),
\tag{26}
$$

*where training data $\mathcal{D}_{n_M}$ and prompt-tuning parameters $\overline{\mathcal{P}}_M$ are variables of the training error $\hat{\mathcal{E}}_M$ of the model in prompt-tuning stage. The number of training samples $n_M$ and the size of parameter space $|\mathcal{P}_M|$ are variables of the generalization gap bound between training error and test error. Moreover, when $n_M$ is fixed, there exist an optimal $|\overline{\mathcal{P}}_M|$ to achieve the lowest upper bound for prompt-tuning-based methods, i.e.,*

$$
\min(\mathcal{U}(\mathcal{E}_M)) = \hat{\mathcal{E}}_M(\mathcal{D}_{n_M}, \overline{\mathcal{P}}_M) + O(\sqrt{|\overline{\mathcal{P}}_M|/n_M}).
\tag{27}
$$

*Proof.* We first derive the generalization bound for existing prompt-tuning-based methods. Based on Lemma C.1, if we ignore the pre-trained model, we can obtain the generalization bound of existing prompt-tuning methods, *i.e.,*

$$\mathcal{U}(\mathcal{E}_M) = \hat{\mathcal{E}}(\mathcal{D}_{n_M}, \mathcal{P}_M) + O(\sqrt{|\mathcal{P}_M|/n_M}). \tag{28}$$

However, training error of the model in the prompt-tuning stage will benefit from the better initialization with the pre-trained stage. Actually, such benefits $B$ on the training error can be written as follows.

$$B = \hat{\mathcal{E}}(\mathcal{D}_{n_M}, \mathcal{P}_M) - \hat{\mathcal{E}}_M(\mathcal{D}_{n_M}, \mathcal{P}_M). \tag{29}$$

Therefore, we can obtain the generalization error bound for existing prompt-tuning-based methods as follows.

$$\begin{aligned}
\mathcal{U}(\mathcal{E}_M) &= \hat{\mathcal{E}}(\mathcal{D}_{n_M}, \mathcal{P}_M) + O(\sqrt{|\mathcal{P}_M|/n_M}) - B \\
&= \hat{\mathcal{E}}(\mathcal{D}_{n_M}, \mathcal{P}_M) + O(\sqrt{|\mathcal{P}_M|/n_M}) - \hat{\mathcal{E}}(\mathcal{D}_{n_M}, \mathcal{P}_M) + \hat{\mathcal{E}}_M(\mathcal{D}_{n_M}, \mathcal{P}_M) \\
&= \hat{\mathcal{E}}_M(\mathcal{D}_{n_M}, \mathcal{P}_M) + O(\sqrt{|\mathcal{P}_M|/n_M}).
\end{aligned} \tag{30}$$

Based on the generalization bound, to examine whether the lowest upper generalization bound exists, we derive the partial derivative of $\mathcal{U}(\mathcal{E}_M)$ with respect to $|\mathcal{P}_M|$, *i.e.,*

$$\begin{aligned}
\frac{\partial(\mathcal{U}(\mathcal{E}_M))}{\partial|\mathcal{P}_M|} &= \frac{\partial(\hat{\mathcal{E}}_M(\mathcal{D}_{n_M}, \mathcal{P}_M))}{\partial|\mathcal{P}_M|} + \frac{\partial O(\sqrt{|\mathcal{P}_M|/n_M})}{\partial|\mathcal{P}_M|} \\
&= \frac{\partial(\hat{\mathcal{E}}_M(\mathcal{D}_{n_M}, \mathcal{P}_M))}{\partial|\mathcal{P}_M|} + O(1/\sqrt{|\mathcal{P}_M| \cdot n_M}).
\end{aligned} \tag{31}$$

In addition, according to the generalization bound, we have the observations as follows. First, with the increase of parameters $\mathcal{P}_M$, a larger parameter size confers stronger optimization ability, leading to the decrease of the training error (*i.e.,* $\hat{\mathcal{E}}_M(\mathcal{D}_{n_M}, \mathcal{P}_M)$) in Eq. (30). In addition, the generalization gap bound (*i.e.,* $O(\sqrt{|\mathcal{P}_M|/n_M})$) increases. Therefore, $\frac{\partial(\hat{\mathcal{E}}_M(\mathcal{D}_{n_M}, \mathcal{P}_M))}{\partial|\mathcal{P}_M|} < 0$ while $O(1/\sqrt{|\mathcal{P}_M| \cdot n_M}) > 0$. As a result, there exist a optimal size of $\mathcal{P}_M$ (*i.e.,* $|\overline{\mathcal{P}}_M|$), which enables the derivative of $\mathcal{U}(\mathcal{E}_M)$ with respect to $|\mathcal{P}_M|$ equals to 0. Therefore, we can achieve the lowest upper bound for prompt-tuning-based methods, *i.e.,*

$$\min(\mathcal{U}(\mathcal{E}_M)) = \hat{\mathcal{E}}_M(\mathcal{D}_{n_M}, \overline{\mathcal{P}}_M) + O(\sqrt{|\overline{\mathcal{P}}_M|/n_M}). \tag{32}$$

Thus we complete the proof. $\qquad\qquad\square$

## C.2    COMPARISON OF THE GENERALIZATION ERROR BOUND

**Theorem C.3.** *(Generalization error bound for the proposed HG-Adapter.) With dual structure-aware adapters and potential labeled data extension, the proposed HG-Adapter decreases both the training error and the generalization gap to achieve a lower generalization error bound $\mathcal{U}(\mathcal{E}_A)_{n_A}$ than that of existing prompt-tuning-based methods (i.e., $\mathcal{U}(\mathcal{E}_M)_{n_M}$), i.e.,*

$$\mathcal{U}(\mathcal{E}_A)_{n_A} < \mathcal{U}(\mathcal{E}_A)_{n_M} < \mathcal{U}(\mathcal{E}_M)_{n_M}, \tag{33}$$

*where $n_A$ indicates the number of training data for the proposed HG-Adapter.*

*Proof.* According to the Theorem 2.3, we have the generalization error bound of existing prompt-tuning-based methods, *i.e.,*

$$\mathcal{U}(\mathcal{E}_M) = \hat{\mathcal{E}}_M(\mathcal{D}_{n_M}, \mathcal{P}_M) + O(\sqrt{|\mathcal{P}_M|/n_M}). \tag{34}$$

Moreover, given the fixed size of the training data $n_M$, we can obtain the lowest generalization error bound when the model achieves the optimal parameters $\overline{\mathcal{P}}_M$, *i.e.,*

$$\min(\mathcal{U}(\mathcal{E}_M)) = \hat{\mathcal{E}}_M(\mathcal{D}_{n_M}, \overline{\mathcal{P}}_M) + O(\sqrt{|\overline{\mathcal{P}}_M|/n_M}). \tag{35}$$

Therefore, when the training data is fixed as $n_M$, we see that the generalization error bound of existing prompt-tuning-based methods follows the U-shaped behavior. That is, as the size of parameters $\mathcal{P}_M$

Table 3: Statistics of all datasets.

| Datasets | #Nodes | #Node Types | #Edges | #Edge Types | Target Node/Edge | #Training | #Test | #Class |
|----------|--------|-------------|--------|-------------|------------------|-----------|-------|--------|
| ACM | 8,994 | 3 | 25,922 | 4 | Paper | 600 | 2,125 | 3 |
| Yelp | 3,913 | 4 | 72,132 | 6 | Bussiness | 300 | 2,014 | 3 |
| DBLP | 18,405 | 3 | 67,946 | 4 | Author | 800 | 2,857 | 4 |
| Aminer | 55,783 | 3 | 153,676 | 4 | Paper | 80 | 1,000 | 4 |
| HBN-B | 15,322 | 4 | 6,135,187 | 10 | Drug-Target | 7,875 | 875 | 2 |
| Ogbn-mag | 1,939,743 | 4 | 36,805,743 | 7 | Paper | 625,930 | 66,275 | 349 |

increases, the generalization error of the model first decreases until the generalization error is lowest at the optimal parameters $\overline{\mathcal{P}}_M$, and then as the size of parameters $\mathcal{P}_M$ increases, the generalization error of the model increases. As we mentioned above, previous methods generally ignore the graph structures and thus may not be sufficient to model the input data and increase the training error $\hat{\mathcal{E}}_M$. As a result, we can obtain that $|\mathcal{P}_M| < |\overline{\mathcal{P}}_M|$. To solve this issue, the proposed method designs dual structure-aware adapters to tune the heterogeneous and homogeneous graph structures by increasing a few parameters. Then the parameters $\mathcal{P}_A$ are expected to be closer to the optimal parameters $\overline{\mathcal{P}}_M$ than the parameters $\mathcal{P}_M$, *i.e.,*

$$||\mathcal{P}_A| - |\overline{\mathcal{P}}_M|| < ||\mathcal{P}_M| - |\overline{\mathcal{P}}_M||. \tag{36}$$

We then have

$$\mathcal{U}(\mathcal{E}_A) - \min(\mathcal{U}(\mathcal{E}_M)) < \mathcal{U}(\mathcal{E}_M) - \min(\mathcal{U}(\mathcal{E}_M)). \tag{37}$$

Then, when the training data is fixed as $n_M$, we can obtain

$$\mathcal{U}(\mathcal{E}_A)_{n_M} < \mathcal{U}(\mathcal{E}_M)_{n_M}. \tag{38}$$

In addition to the dual adapters, we further conduct the potential labeled data extension by designing the label-propagated contrastive loss and two self-supervised losses, thus incorporating all unlabeled nodes as supervision signals. Therefore, the size of training data after extension (*i.e.,* $n_A$) is supposed to be larger than the original size of training data $n_M$. Therefore, we have

$$O(\sqrt{|\mathcal{P}_A|/n_A}) < O(\sqrt{|\mathcal{P}_A|/n_M}). \tag{39}$$

We then have

$$\mathcal{U}(\mathcal{E}_A)_{n_A} < \mathcal{U}(\mathcal{E}_A)_{n_M} < \mathcal{U}(\mathcal{E}_M)_{n_M}. \tag{40}$$

Thus we complete the proof. □

## D EXPERIMENTAL SETTINGS

This section provides detailed experimental settings in Section Experiments, including the description of all datasets in Section D.1, summarization of all comparison methods in Section D.2, evaluation protocol in Section D.3, model architectures and settings in Section D.4, and computing resource details in Section D.5.

### D.1 DATASETS

We use four public heterogeneous graph datasets from various domains including three academic datasets (*i.e.,* ACM (Wang et al., 2019), DBLP (Wang et al., 2019), and Aminer (Hu et al., 2019)), and one business dataset (*i.e.,* Yelp (Zhao et al., 2021)). Table 3 summarizes the data statistics. We list the details of the datasets as follows.

- **ACM** is an academic heterogeneous graph dataset. It contains three types of nodes (paper (P), author (A), subject (S)), four types of edges (PA, AP, PS, SP), and treats categories of papers as labels.

- **Yelp** is a business heterogeneous graph dataset. It contains three types of nodes (business (B), user (U), service (S), level (L)), six types of edges (BU, UB, BS, SB, BL, LB), and treats categories of businesses as labels.

- **DBLP** is an academic heterogeneous graph dataset. It contains three types of nodes (paper (P), authors (A), conference (C)), four types of edges (PA, AP, PC, CP), and treats research areas of authors as labels.

- **Aminer** is an academic heterogeneous graph dataset. It contains three types of nodes (paper (P), author (A), reference (R)), four types of edges (PA, AP, PR, RP), and treats categories of papers as labels.

- **HBN-B** is a biomedical heterogeneous graph dataset. It contains four types of nodes (drug (Dr), target (T), disease (Di)), side-effect (S), ten types of edges (DrT, DrDr, DrDi, DrS, TT, TDi, TDr, DiDr, SDr, DiT), and treats the existence of drug-target interaction as labels.

- **Ogbn-mag** is a large-scale academic heterogeneous graph dataset. It contains four types of nodes (paper (P), Author (A), Institution (I), Field (F)), seven types of edges (PA, PP, PF, AI, AP, FP, IA), and treats categories of papers as labels.

Table 4: The characteristics of all comparison methods.

| Methods | Semi-supervised | Fine-tuning | Prompt-tuning | Adapter-tuning |
|---------|:---------------:|:-----------:|:-------------:|:--------------:|
| HAN (2019) | ✓ | | | |
| HGT (2020) | ✓ | | | |
| DMGI (2020) | | ✓ | | |
| HDMI (2021) | | ✓ | | |
| HeCo (2021) | | ✓ | | |
| HGCML (2023) | | ✓ | | |
| HGMAE (2023) | | ✓ | | |
| HERO (2024) | | ✓ | | |
| HGPrompt (2024) | | | ✓ | |
| HetGPT (2024) | | | ✓ | |
| HG-Adapter (ours) | | | | ✓ |

## D.2 COMPARISON METHODS

The comparison methods include two traditional semi-supervised methods (*i.e.,* HAN (Wang et al., 2019) and HGT (Hu et al., 2020b)), six fine-tuning-based methods (*i.e.,* DMGI (Park et al., 2020), HDMI (Jing et al., 2021), HeCo (Wang et al., 2021), HGCML (Wang et al., 2023), HGMAE (Tian et al., 2023), and HERO (Mo et al., 2024)), and two prompt-tuning-based methods (*i.e.,* HGPrompt (Yu et al., 2024a) and HetGPT (Ma et al., 2024)), where the pre-training and prompt-tuning of HGPrompt are specifically designed, while HetGPT only designs the prompt-tuning and thus can be used for different pre-trained models. The characteristics of all methods are listed in Table 4, where "Semi-supervised", "Fine-tuning", "Prompt-tuning", and "Adapter-tuning" indicate that the method conducts semi-supervised learning, fine-tuning, prompt-tuning, and adapter-tuning, respectively.

## D.3 EVALUATION PROTOCOL

We follow the evaluation protocol in previous works (Jing et al., 2021; Pan & Kang, 2021; Zhou et al., 2022) to conduct node classification and node clustering as semi-supervised and unsupervised downstream tasks, respectively.

For fine-tuning-based methods, we first train models with unlabeled data in a self-supervised manner and output learned node representations. After that, the resulting representations can be used for different downstream tasks. For the node classification task, we train a simple logistic regression classifier with a fixed iteration number, and then evaluate the effectiveness with Micro-F1 and Macro-F1 scores. For the node clustering task, we conduct clustering and split the learned representations into $c$ clusters with the K-means algorithm, then calculate the normalized mutual information (NMI) and average rand index (ARI) to evaluate the performance of node clustering.

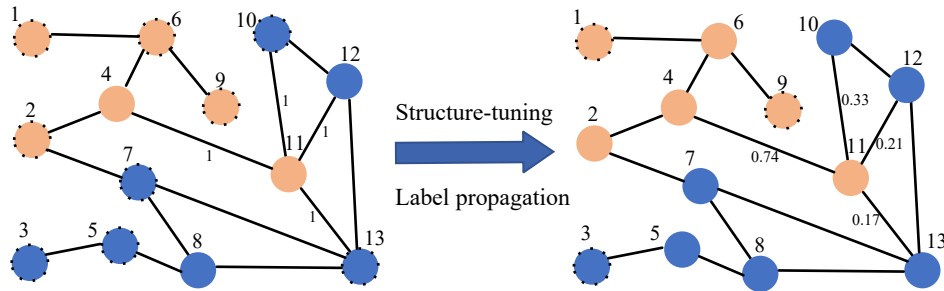

Figure 3: Toy example sampled from the ACM dataset, where blue nodes belong to class 1 and yellow nodes belong to class 2. The nodes with dashed lines are unlabeled nodes in the test set, and the nodes without dashed lines are labeled nodes in the training set. The left side is the original graph with edge weights of 1, and the right side is the graph after adapter-tuning, where labeled nodes are extended and edge weights are tuned.

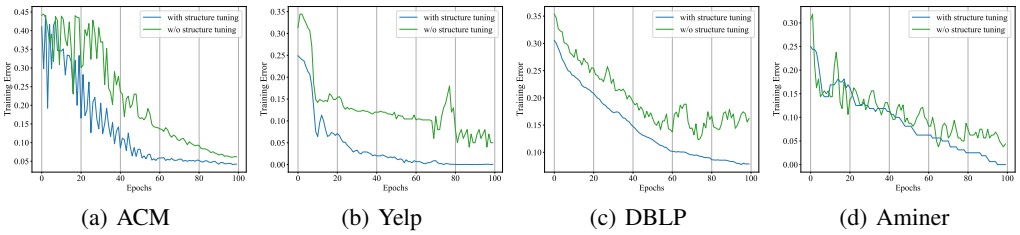

Figure 4: The training error of the proposed method with and without the structure tuning on all heterogeneous graph datasets.

For prompt-tuning-based methods and the proposed adapter-tuning-based method, the models are also pre-trained in an unsupervised manner and output frozen representations. In the tuning stage, for the node classification task, we first obtain the predicted probability of each node by calculating the similarity between the prediction vector and each class-subgraph representation, and then using the softmax function to obtain class probabilities. Then, the class with the maximum likelihood for each node is designated as the predicted class. We also evaluate the effectiveness of models with Micro-F1 and Macro-F1 scores. For the node clustering task, we input the class probabilities to the K-means algorithm, then calculate the normalized mutual information (NMI) and average rand index (ARI) to evaluate the performance of node clustering.

### D.4 MODEL ARCHITECTURES AND SETTINGS

As described in Section 2, the proposed method employs homogeneous and heterogeneous adapters to tune the homogeneous and heterogeneous graph structures (*i.e.,* $\mathbf{A}$ and $\mathbf{S}$) and capture task-related structural information. Moreover, the proposed method employs the projection (*i.e.,* $g_\vartheta$) to obtain the prediction matrix $\mathbf{P}$. In the proposed method, homogeneous and heterogeneous adapters are simply implemented by two linear layers, followed by the ReLU activation. In addition, the projection $g_\vartheta$ is also simply implemented by the linear layer. Finally, In the proposed method, all parameters were optimized by the Adam optimizer (Kingma & Ba, 2015) with an initial learning rate. In all experiments, we repeat the experiments five times for all methods and report the average results.

### D.5 COMPUTING RESOURCE DETAILS

All experiments were implemented in PyTorch and conducted on a server with 8 NVIDIA GeForce 3090 (24GB memory each). Almost every experiment can be done on an individual 3090, and the training time of all comparison methods as well as our method, is less than 1 hour.

Table 5: Performance on biomedical and large-scale heterogeneous graph datasets, where the best results are highlighted in bold, while improved results with the proposed HG-Adapter are underlined. The "+" symbol indicates the integration of HG-Adapter and HetGPT with original pre-trained HGNN models. "OOM" indicates out-of-memory, "-" indicates that the method is not suitable for the task, and ∗ indicates the mini-batch version method on the Ognb-mag dataset.

| Method | HBN-B | | Ogbn-mag |
|---|---|---|---|
| | AUC | AUPR | Accuracy |
| HAN | - | - | OOM |
| HGT | 84.9±0.6 | 81.3±0.5 | 49.8±0.9 |
| DMGI | - | - | OOM |
| HGCML | - | - | OOM |
| HGMAE | 86.3±0.6 | 87.6±0.2 | OOM |
| HGPrompt | 86.6±0.5 | 87.8±0.4 | OOM |
| HDMI∗ | - | - | 46.6±0.7 |
| +HetGPT | - | - | 47.1±0.6 |
| +HG-Adapter | - | - | 49.5±0.5 |
| HeCo∗ | 86.1±0.4 | 87.5±0.7 | 48.7±0.5 |
| +HetGPT | 86.7±0.6 | 88.1±0.8 | 49.3±0.8 |
| +HG-Adapter | **87.7±0.5** | **88.9±0.6** | 50.5±0.7 |
| HERO∗ | 85.6±0.7 | 86.8±0.5 | 49.5±0.4 |
| +HetGPT | 86.0±0.6 | 87.1±0.8 | 50.6±0.3 |
| +HG-Adapter | 87.3±0.5 | 88.5±0.7 | **52.1±0.5** |

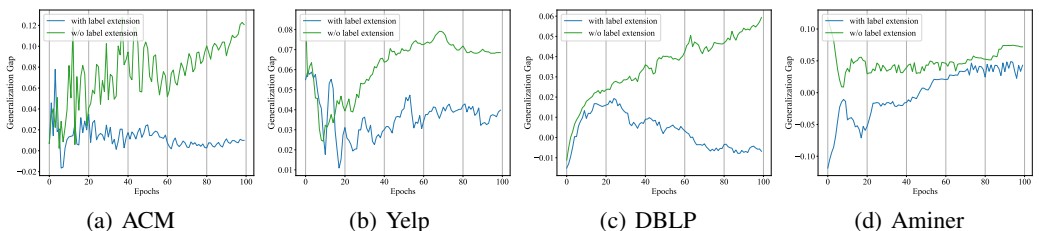

(a) ACM      (b) Yelp      (c) DBLP      (d) Aminer

Figure 5: The generalization gap (the difference between test error and training error) of the proposed method with and without the labeled data extension on all heterogeneous graph datasets.

# E   ADDITIONAL EXPERIMENTS

This section provides some additional experimental results to support the proposed method, including the motivation illustration in Section E.1, the experimental results on biomedical and large-scale datasets in Section E.2, the ablation study on the structure tuning in Section E.3, the ablation study on the potential labeled data extension in Section E.4, the ablation study on the parameters in the proposed method in Section E.5, the ablation study on the dimensions of dual adapters in Section E.6, the ablation study on dual adapters in Section E.7, the ablation study of the margin loss in Section E.8, visualization of the learned representations in Section E.9, parameter analysis in Section E.10, experimental results on the node clustering task in Table 8.

## E.1   MOTIVATION ILLUSTRATION

In this paper, we point out that existing methods only focus on node features while ignoring graph structures, thereby increasing the training error. Moreover, existing methods may be constrained by the limited labeled data during the prompt-tuning stage, leading to a large generalization gap. To further explain above motivation, we sample a few nodes from the ACM dataset and construct a toy example and implement adapter-tuning and label propagation on it, and visualize them in Figure 3, where different colors indicate nodes from different classes, nodes with dashed lines are unlabeled nodes in the test set, and the nodes without dashed lines are labeled nodes in the training set.

Table 6: Classification performance (*i.e.,* Macro-F1 and Micro-F1) of the variant methods without homogeneous and heterogeneous adapters on all heterogeneous graph datasets.

| Method | ACM | | Yelp | | DBLP | | Aminer | |
|---|---|---|---|---|---|---|---|---|
| | Macro-F1 | Micro-F1 | Macro-F1 | Micro-F1 | Macro-F1 | Micro-F1 | Macro-F1 | Micro-F1 |
| w/o hom-adapter | 92.1±0.6 | 92.0±0.5 | 91.1±0.7 | 90.7±0.8 | 83.7±1.2 | 84.5±1.0 | 73.2±0.8 | 82.3±0.9 |
| w/o het-adapter | 86.0±0.7 | 85.3±0.6 | 92.2±0.8 | 91.4±0.7 | 81.4±0.7 | 82.5±0.6 | 71.9±0.5 | 80.1±0.3 |
| Proposed | **92.7±0.4** | **92.7±0.7** | **93.1±0.6** | **92.7±0.5** | **94.0±0.7** | **94.7±0.8** | **78.3±0.5** | **87.1±0.6** |

Table 7: Classification performance (*i.e.,* Macro-F1 and Micro-F1) of the variant methods with the proposed margin loss and the InfoNCE loss on all heterogeneous graph datasets.

| Method | ACM | | Yelp | | DBLP | | Aminer | |
|---|---|---|---|---|---|---|---|---|
| | Macro-F1 | Micro-F1 | Macro-F1 | Micro-F1 | Macro-F1 | Micro-F1 | Macro-F1 | Micro-F1 |
| InfoNCE loss | 89.6±0.9 | 89.5±0.7 | 91.1±0.5 | 91.4±0.6 | 92.1±0.8 | 93.0±0.7 | 75.8±0.4 | 82.6±0.5 |
| Margin loss | **92.7±0.4** | **92.7±0.7** | **93.1±0.6** | **92.7±0.5** | **94.0±0.7** | **94.7±0.8** | **78.3±0.5** | **87.1±0.6** |

From Figure 3, we have the observations as follows. First, if we only tune the node features and ignore tuning the graph structures, some nodes in the training set may be misclassified, thereby increasing the training error. For example, for node 11, it will aggregate much information from the nodes (nodes 10, 12, 13) of another class after the message-passing with the original graph structures. Therefore, this may cause nodes to confuse their own class information, thus increasing the training error. In contrast, if we tune both the node features and ignore tuning the graph structures, the misclassified node 11 may be corrected by re-weighting the edge weight. As a result, the proposed method decreases the training error thus improve the model generalization. Second, compared with unlabeled nodes, the ratio of labeled nodes is very small, result in a large generalization gap between the training error and the test error. However, after the label propagation, the number of labeled nodes increases greatly. As a result, the proposed method decreases the generalization gap thus improve the model generalization.

### E.2 EFFECTIVENESS ON DATASET FROM OTHER DOMAINS AND LARGE-SCALE DATASET

To further verify the model's generalization ability across different domains, we evaluate the proposed method on the biomedical heterogeneous graph dataset HBN-B (Li et al., 2022) and the large-scale heterogeneous graph dataset (i.e., Ogbn-mag (Hu et al., 2020a)), and report the results in Table 5.

Obviously, on the biomedical heterogeneous graph dataset HBN-B, the proposed method consistently obtains improvements on the pre-trained HGNNs (i.e., HeCo and HERO). For instance, the proposed method on average, improves by 1.3%, compared to the baseline method HeCo in terms of AUC and AUPR. In addition, the proposed HG-Adapter also obtains significant improvements to the prompt-tuning method. For instance, the proposed method on average, improves by 1.3%, compared to the best prompt-tuning method (i.e., HGPrompt) in terms of AUC and AUPR. Therefore, the effectiveness and generalization ability of the proposed method is further verified on datasets from different domains. In addition, on the large-scale heterogeneous graph dataset Ogbn-mag, the proposed method always obtains promising results, compared to the original baselines (HDMI, HeCo, and HERO) as well as the prompt-tuning-based method (i.e., HetGPT). For example, the proposed method improves by 5.3% and 3.0%, compared to the baseline method HERO and prompt-tuning-based method HetGPT, respectively, in terms of Accuracy. Therefore, the effectiveness and scalability of the proposed method are further verified.

### E.3 EFFECTIVENESS OF THE STRUCTURE TUNING

The proposed method designs the dual adapters to capture more task-related structural information than prompt-tuning-based methods. To verify the effectiveness of the structure tuning with dual adapters, we investigate the training error of the proposed method with and without the structure tuning, and report the results in Figure 4. Obviously, as the number of epochs increases, the training

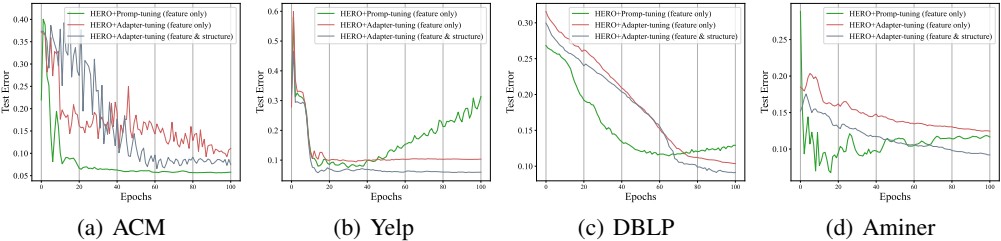

(a) ACM (b) Yelp (c) DBLP (d) Aminer

Figure 6: The test errors of the HERO with prompt-tuning on features, adapter-tuning on features, and adapter-tuning on both features and structures on all heterogeneous graph datasets.

error of the proposed method with structure tuning continues to decrease and finally tends to 0. This indicates that the proposed method fits the input data well. Moreover, the proposed method with structure tuning obtains consistently lower training error than the method without structure tuning. The reason can be attributed to the fact that the structure tuning enables the model to fit the input data better and get closer to the optimal parameters $\overline{\mathcal{P}}_M$, thus reducing the training error and improving the model's generalization ability.

### E.4 EFFECTIVENESS OF THE POTENTIAL LABELED DATA EXTENSION

The proposed method designs the potential labeled data extension to incorporate both labeled and unlabeled nodes as supervision signals, thus alleviating the limited labeled data in the tuning stage. To verify the effectiveness of the potential labeled data extension, we investigate the generalization gap of the proposed method with and without the label extension and report the results in Figure 5. From Figure 5, we can find that the proposed method with the potential labeled data extension consistently achieves a smaller generalization gap than the method without label extension. This is reasonable because the label extension increases the number of training samples potentially, thus decreasing the generalization gap bound $O(\sqrt{|\overline{\mathcal{P}}_M|/n_M})$ and further decreasing the generalization error bound of existing methods.

### E.5 EFFECTIVENESS OF THE PARAMETERS IN THE PROPOSED METHOD

The proposed method designs dual adapters with parameters $|P_A|$ to fit better the input data to decrease the training error, thus decreasing the upper bound of the test error and approaching the optimal parameters $|\bar{P}_M|$. To verify that $|P_A|$ is indeed closer to the optimal parameters $|\bar{P}_M|$ than existing prompt-tuning-based methods, we remove the label extension module and fix the number of training samples, and then implement several variant methods (i.e., adapter-tuning on both node features and graph structures, adapter-tuning on only node features, and prompt-tuning on only node features), and report the results in Figure 6. Obviously, when the number of training samples is fixed, the proposed adapter-tuning on node features and graph structures always obtains lower test error than the prompt-tuning and adapter-tuning on only node features. As a result, we can obtain that the parameters $|P_A|$ of the proposed adapter-tuning is indeed closer to $|\bar{P}_M|$ than existing prompt-tuning-based methods.

### E.6 ABLATION STUDY OF THE DIMENSION OF DUAL-ADAPTERS

Theorem 2.3 indicates that the upper bound of the test error exhibits a U-shaped pattern, where it initially decreases and then increases as the number of parameters grows. Therefore, we cannot further improve the performance by simply increasing the size of dual adapters. To verify it, we conduct the ablation study by varying the size of each adapter, and report the results in Figure 7. From Figure 7, we can find that that as the adapter size increases, the performance of the model may first improve, and then decrease when the size is too large. This is consistent with our theoretical results above, i.e., as the parameter size increases, the upper bound of the test error decreases first and then increases. Correspondingly, the performance of the model may increase first and then decrease.

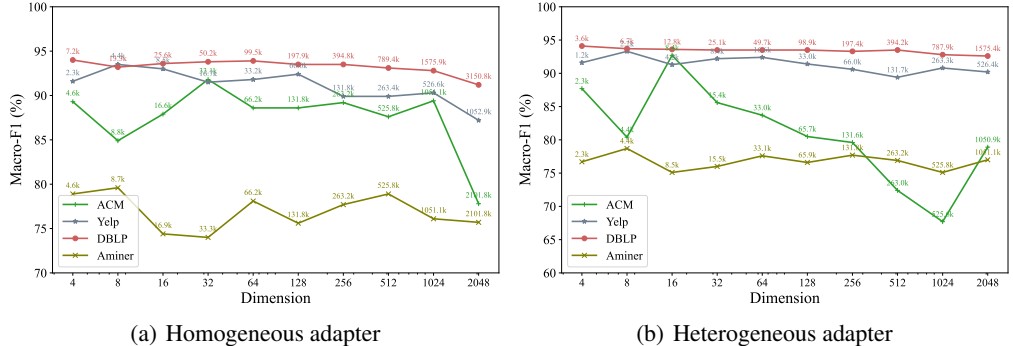

(a) Homogeneous adapter          (b) Heterogeneous adapter

Figure 7: The performance of the proposed method under different adapter dimensions and corresponding parameter sizes.

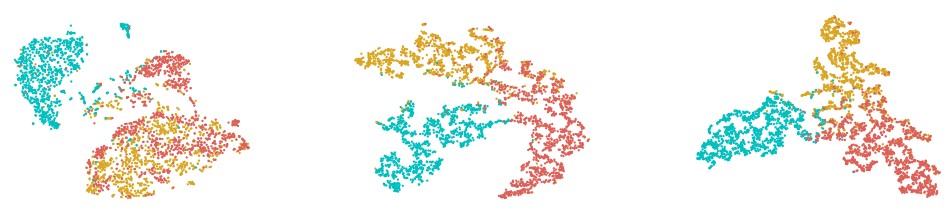

(a) HERO+Fine-Tuning (SIL: 0.31)    (b) HERO+HetGPT (SIL: 0.38)    (c) HERO+HG-Adapter (SIL: 0.44)

Figure 8: Visualization plotted by t-SNE and the corresponding silhouette scores (SIL) of the representations learned by HERO with fine-tuning, prompt-tuning (*i.e.,* HetGPT), and the proposed adapter-tuning on the DBLP dataset, respectively.

### E.7 ABLATION STUDY OF DUAL ADAPTERS

The proposed method designs dual adapters to adaptively tune the homogeneous and heterogeneous graph structures (*i.e.,* **A** and **S**) to capture the task-related information. We demonstrate the effectiveness of dual adapters in Section 3.2.2. To further verify it, we investigate the node classification performance of the variants methods without the homogeneous and heterogeneous adapters, respectively, and report the results in Table 6. Obviously, the proposed method with dual adapters obtains superior performance than the variant methods without homogeneous and heterogeneous adapters. The reason can be attributed to the fact that dual adapters tunes homogeneous and heterogeneous graph structures simultaneously, thus can better fit the input data than variant method without adapters to achieve lower generalization error bound and better performance on different downstream tasks. As a result, the effectiveness of dual adapters is verified again.

### E.8 ABLATION STUDY OF THE MARGIN LOSS

The proposed method designs the margin loss to optimize the heterogeneous graph structure **S** and treat both labeled and unlabeled nodes as equal supervision signals. The margin loss aims to decrease the distance $d(\mathbf{c}_{\tilde{\mathbf{y}}_i}, \hat{\mathbf{m}}_i)^2$, while increasing the distance $d(\mathbf{c}_{\tilde{\mathbf{y}}_i}, \hat{\mathbf{m}}_j)^2$, thus satisfying the "safe" distance between them. To verify the effectiveness of the margin loss, we investigate the node classification performance of the variants method with the InfoNCE loss instead of the proposed margin loss and report the results in Table 7. From Table 7, we can find that the proposed method with the margin loss obtains better performance than the variant method with the InfoNCE loss. This can be attributed to the fact that directly aligning class-subgraph representation and adapted representation is unreasonable since they come from different feature distributions.

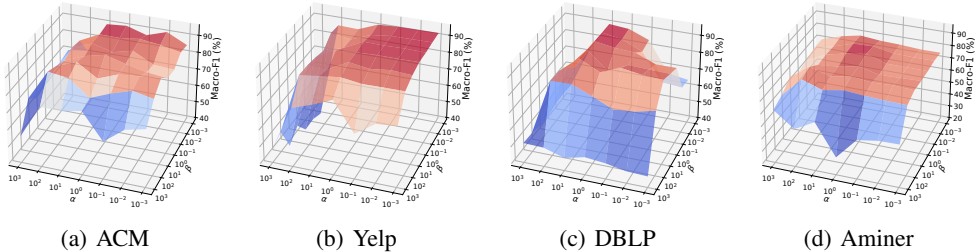

|(a) ACM|(b) Yelp|(c) DBLP|(d) Aminer|

Figure 9: The classification performance of the proposed method at different parameter settings (*i.e.*, $\alpha$ and $\beta$) on all heterogeneous graph datasets.

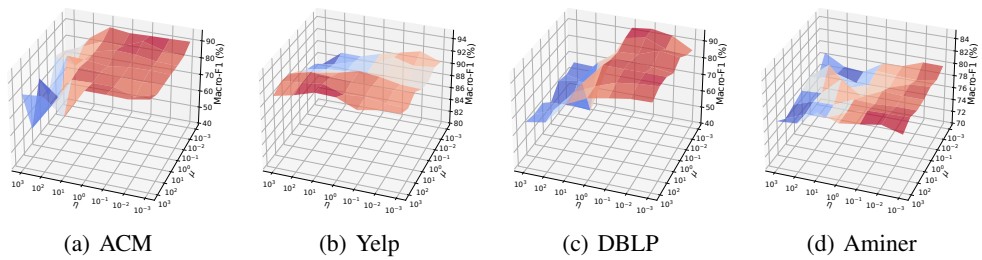

|(a) ACM|(b) Yelp|(c) DBLP|(d) Aminer|

Figure 10: The classification performance of the proposed method at different parameter settings (*i.e.*, $\eta$ and $\mu$) on all heterogeneous graph datasets.

### E.9 VISUALIZATION OF LEARNED REPRESENTATIONS

To further verify the effectiveness of the learned representations, we visualize node representations learned by the combinations of pre-trained HERO and different tuning methods (fine-tuning, prompt-tuning, and proposed adapter-tuning) on the ACM dataset and report the results and corresponding silhouette scores (SIL) in Figure 8. Obviously, in the visualization, the node representations learned by the proposed HG-Adapter exhibit better clustering status, *i.e.,* nodes with different class labels are more widely separated. Moreover, the representations learned by the proposed method obtain the best silhouette score, compared to other methods (*i.e.,* fine-tuning, and HetGPT). The reason can be attributed to the fact that the proposed method designs a contrastive loss based on class subgraph similarity to ensure that nodes within the same class are close to each other, thereby enhancing clustering performance.

### E.10 PARAMETER ANALYSIS

In the proposed method, we employ non-negative parameters $\alpha$ and $\beta$ to achieve a trade-off between the adapted representations and the frozen representations. Moreover, we employ non-negative parameters $\eta$ and $\mu$ between different terms of the final objective function $\mathcal{J}$. To investigate the impact of $\alpha$ and $\beta$ as well as $\eta$ and $\mu$ with different settings, we conduct the node classification on all heterogeneous graph datasets by varying the value of parameters in the range of $[10^{-3}, 10^3]$ and reporting the results in Figure 9 and Figure 10.

From Figure 9, we can find that if the values of parameters (*i.e.,* $\alpha$ and $\beta$) are too large (*e.g.,* $10^3$), the proposed method cannot achieve satisfactory performance. This verifies that both the adapted representations and the frozen representations are important for downstream tasks. In addition, from Figure 10, we can also find that if the values of parameters (*i.e.,* $\eta$ and $\mu$) are too large (*e.g.,* $10^3$), the proposed method obtains inferior performance. The reason can be attributed to the fact that when $\eta$ and $\mu$ are large, the effect of the contrastive loss $\mathcal{L}_{con}$ may be affected, thus failing to provide sufficient label guidance for the model.

Table 8: Clustering performance (*i.e.,* NMI and ARI) on all heterogeneous graph datasets, where the best results are highlighted in bold, while improved results with the proposed HG-Adapter are underlined. The "+" symbol indicates the integration of HG-Adapter and HetGPT with original pre-trained HGNN models.

| Method | ACM | | Yelp | | DBLP | | Aminer | |
|--------|-----|-----|------|-----|------|-----|--------|-----|
| | NMI | ARI | NMI | ARI | NMI | ARI | NMI | ARI |
| HAN | 65.6±1.3 | 67.4±1.5 | 37.8±0.9 | 40.1±1.1 | 77.5±0.7 | 83.0±0.8 | 35.5±0.6 | 31.6±0.5 |
| HGT | 68.9±0.9 | 69.9±0.8 | 39.1±0.6 | 41.2±0.7 | **78.6±0.4** | **83.9±0.6** | 36.1±0.6 | 32.3±0.8 |
| DMGI | 67.8±0.9 | 70.2±1.0 | 36.8±0.6 | 34.4±0.7 | 72.2±0.8 | 72.8±0.9 | 27.3±0.9 | 23.1±0.8 |
| HGCML | 69.1±0.7 | 71.6±0.8 | 37.4±0.6 | 39.5±0.8 | 74.5±0.9 | 75.1±1.1 | 35.9±0.6 | 31.1±0.5 |
| HGMAE | 69.7±0.8 | 72.6±0.6 | 40.3±0.9 | 42.4±0.8 | 76.9±0.6 | 82.3±0.7 | 41.1±0.8 | 38.3±0.9 |
| HGPrompt | 69.2±0.4 | 72.0±0.5 | 37.5±0.4 | 39.7±0.7 | 76.1±0.6 | 81.2±0.8 | 37.2±0.9 | 33.8±1.1 |
| HDMI | 69.5±0.5 | 72.3±0.7 | 38.9±0.6 | 40.7±0.8 | 73.1±0.3 | 74.4±0.4 | 33.5±0.4 | 28.9±0.5 |
| +HetGPT | 70.1±0.6 | 72.8±0.8 | 39.2±0.6 | 41.5±0.7 | 74.8±0.9 | 75.7±1.1 | 34.4±0.7 | 29.3±0.6 |
| +HG-Adpater | 70.4±0.7 | 73.1±0.6 | 39.7±0.5 | 42.2±0.4 | 75.9±0.6 | 80.7±0.8 | 35.1±0.9 | 30.9±1.0 |
| HeCo | 67.8±0.8 | 70.5±0.7 | 39.3±0.6 | 42.1±0.8 | 74.5±0.8 | 80.1±0.9 | 32.2±1.1 | 28.6±1.0 |
| +HetGPT | 68.2±0.6 | 70.8±0.5 | 40.1±0.7 | 42.5±0.9 | 75.1±0.5 | 80.5±0.7 | 32.7±0.4 | 29.2±0.6 |
| +HG-Adpater | 69.0±0.4 | 72.9±0.3 | **41.2±0.5** | **43.1±0.6** | 76.3±0.9 | 81.9±1.0 | 34.1±0.7 | 30.3±0.8 |
| HERO | 68.8±0.6 | 71.8±0.6 | 38.6±0.8 | 40.6±0.9 | 74.1±0.7 | 79.3±0.7 | 36.8±0.7 | 35.3±0.9 |
| +HetGPT | 69.7±0.5 | 72.3±0.4 | 39.3±0.7 | 41.3±0.9 | 74.6±0.6 | 80.4±0.5 | 40.2±0.6 | 37.1±0.7 |
| +HG-Adapter | **70.5±0.8** | **73.3±0.9** | 41.1±0.6 | 42.8±0.5 | 76.2±0.8 | 81.4±0.7 | **42.3±0.4** | **39.4±0.5** |

# F  POTENTIAL LIMITATIONS AND FUTURE DIRECTIONS

In this paper, we design dual adapters to capture task-related structural information, thus approaching the optimal parameters and benefiting the model's generalization ability. Moreover, we design contrastive loss, feature reconstruction loss, and margin loss to optimize dual adapters as well as extend potential labeled data. Actually, the feature reconstruction loss relies on the feature-label consistency assumption. That is, if two nodes have the same label, their node features will be similar. However, there are a few cases in which nodes have the same label but totally different node features. In that case, the learned graph structure may suffer from noisy connections. In addition, the quality of the propagated labels of unlabeled data will be affected correspondingly because it relies on the learned graph structure. To solve these issues, we may conduct class feature reconstruction instead of node feature reconstruction to address this issue. We consider these aspects as potential directions for future research.

