# OpenReview forum: "HG-Adapter: Improving Pre-Trained Heterogeneous Graph Neural Networks with Dual Adapters"
_ICLR.cc/2025/Conference — ICLR 2025 Poster_

### Official Review · Reviewer_PQvi · 2024-10-28

**Soundness:** 2
**Presentation:** 3
**Contribution:** 2
**Rating:** 6
**Confidence:** 3

**Summary:**

The paper "HG-Adapter: Improving Pre-Trained Heterogeneous Graph Neural Networks with Dual Adapters" introduces a new framework to enhance the generalization of pre-trained heterogeneous graph neural networks (HGNNs). It addresses two key challenges: insufficient focus on graph structures during tuning and a lack of labeled data, leading to a generalization gap. The proposed HG-Adapter employs dual adapters to capture both homogeneous and heterogeneous graph patterns, improving task-specific performance. It also incorporates a label-propagated contrastive loss and self-supervised learning to utilize unlabeled data, effectively expanding the labeled dataset. This helps reduce the generalization error between training and testing phases.

**Strengths:**

1. The paper introduces a novel approach by using dual adapters to capture both homogeneous and heterogeneous graph structures. This allows for better adaptation to specific graph patterns, improving the model’s generalization capabilities across diverse downstream tasks.

2. By incorporating a label-propagated contrastive loss and self-supervised learning, the paper effectively leverages unlabeled data to extend the training dataset. This approach helps overcome the limitation of scarce labeled data, which is a common challenge in heterogeneous graph neural network applications.

3. The paper provides a solid theoretical foundation by deriving generalization error bounds for prompt-tuning methods. Additionally, it validates the proposed HG-Adapter through extensive experiments, demonstrating superior performance compared to state-of-the-art fine-tuning and prompt-tuning techniques across multiple datasets.

**Weaknesses:**

1. While the proposed HG-Adapter framework is innovative, its dual adapter system and the integration of multiple losses (label-propagated contrastive loss and self-supervised learning) might make the model computationally expensive and challenging to implement in real-world settings with limited resources.

2. The experiments in the paper, while comprehensive, may be limited in the diversity of graph types evaluated. A more extensive validation across different types of heterogeneous graphs, particularly in domains beyond those tested (e.g., biological networks, industrial applications), would provide stronger evidence of the model's generalization ability.

3. Although the framework shows improved performance, its scalability to very large datasets or extremely large graphs is not thoroughly addressed. Given that graph neural networks are often used in scenarios involving large-scale data, the paper could benefit from further analysis on how the model performs with increasing graph size and complexity.

**Questions:**

please see the weaknesses

---

> ### Author Response · Authors · 2024-11-22
> **Response to Reviewer PQvi**
>
> Thanks for the positive comments on the novelty, and experimental results of our method. We are so encouraged and will try our best to address the concerns one by one.
>
> > **Q1.** While the proposed HG-Adapter framework is innovative, its dual adapter system and the integration of multiple losses (label-propagated contrastive loss and self-supervised learning) might make the model computationally expensive and challenging to implement in real-world settings with limited resources.
>
> **A1.** In the revision, we analyzed the time complexity of the proposed HG-Adapter to show its efficiency as follows.
>
> HG-Adapter consists of two prats, i.e., dual structure-aware adapters and potential labeled data extension. We analyze the time complexity of each part as follows.
>
> First, the time complexity of the dual structure-aware adapters is $\mathcal{O}(nkd + n|\mathcal{R}|)$, where $n$, $k$, $d$, and $|\mathcal{R}|$ indicate the number of nodes, the number of neighbors of each node, the number of representation dimensions, and the number of edge types, respectively. Second, the time complexity of the potential labeled data extension is $\mathcal{O}(nkc + nc^2 + nkf)$, where $c$ and $f$ indicate the number of classes and dimensions of node features, respectively. Therefore, the overall time complexity of the proposed HG-Adapter is $\mathcal{O}(n(kd + |\mathcal{R}| + kc + c^2 + kf))$. As a result, The proposed HG-Adapter is scaled linearly with the sample size and has the potential to be implemented with limited resources.
>
> > **Q2.** The experiments in the paper, while comprehensive, may be limited in the diversity of graph types evaluated. A more extensive validation across different types of heterogeneous graphs, particularly in domains beyond those tested (e.g., biological networks, industrial applications), would provide stronger evidence of the model's generalization ability.
>
> **A2.** Thanks for your suggestion. In our original submission, we evaluated the proposed method with three academic datasets and one business dataset. In the revision, to further verify the model's generalization ability across different domains, we evaluate the proposed method on the biomedical heterogeneous graph dataset HBN-B [1], and report the results in Table 5 in Appendix E. Obviously, the proposed method consistently obtains improvements on the pre-trained HGNNs (i.e., HeCo and HERO). For instance, the proposed method on average, improves by 1.3%, compared to the baseline method HeCo in terms of AUC and AUPR. In addition, the proposed HG-Adapter also obtains significant improvements to the prompt-tuning method. For instance, the proposed method on average, improves by 1.3%, compared to the best prompt-tuning method (i.e., HGPrompt) in terms of AUC and AUPR. Therefore, the effectiveness and generalization ability of the proposed method is further verified on datasets from different domains.
>
> [1] Heterogeneous Graph Attention Network for Drug-Target Interaction Prediction. In CIKM 2022.
>
> > **Q3.** Although the framework shows improved performance, its scalability to very large datasets or extremely large graphs is not thoroughly addressed. Given that graph neural networks are often used in scenarios involving large-scale data, the paper could benefit from further analysis on how the model performs with increasing graph size and complexity.
>
> **A3.** Thanks for your suggestion. Based on our complexity analysis in **A1**, the proposed method is scaled linearly with the
> sample size and has the potential to be implemented on large-scale datasets. In the revision, to further verify the scalability of the proposed method, we evaluate the proposed method on the large-scale heterogeneous graph dataset with millions of nodes (i.e., Ogbn-mag [2]), and report the results in Table 5 in Appendix E. From Table 5, the proposed method always obtains promising results, compared to the original baselines (i.e., HDMI, HeCo, and HERO) as well as the prompt-tuning-based method (i.e., HetGPT). For example, the proposed method improves by 5.3% and 3.0%, compared to the baseline method HERO and prompt-tuning-based method HetGPT, respectively, in terms of Accuracy. Therefore, the effectiveness and scalability of the proposed method are further verified.
>
> [2] Open Graph Benchmark: Datasets for Machine Learning on Graphs. In NeurIPS 2020.

---

> > ### Comment · Reviewer_PQvi · 2024-11-25
> > **To authors**
> >
> > Thanks for you careful responses, and I would like to update my score to 6.

---

### Official Review · Reviewer_J7pK · 2024-11-03

**Soundness:** 3
**Presentation:** 3
**Contribution:** 4
**Rating:** 8
**Confidence:** 5

**Summary:**

Summary of the paper

The paper pointed out the major limitations remaining currently in the "pre-train, prompt-tuning" paradigm for heterogeneous graph neural networks (HGNNs), which are :the insufficient adaptation of graph structures during prompt-tuning and the challenges posed by limited labeled data. To mitigate these issues, the authors derived the generalization error bound for existing prompt tuning-based methods, and then proposed a unified framework that combines two new adapters with potential labeled data extension to improve the generalization of pre-trained HGNN models.This novel approach aims to improve the generalization capabilities of pre-trained HGNNs across various downstream tasks.

Contributions
+ Designing dual structure-aware adapters to capture task-related homogeneous and heterogeneous structural information. Moreover, we design a label-propagated contrastive loss and two self-supervised losses to achieve the potential labeled data extension.
+ Deriving a unified generalization error bound for existing methods based on the training error and the generalization gap. Moreover, we demonstrate that the proposed method achieves a lower generalization error bound than existing prompt-tuning-based methods to improve the generalization ability of pre-trained HGNN models.
+ Validating the superior effectiveness and generalization of the proposed HG Adapter compared to state-of-the-art fine-tuning-based and prompt-tuning-based methods, demonstrating its adaptability to different pre-trained HGNN models by experiments.

Merits
+ Significant shortcomings in existing prompt-tuning approaches were pointed clearly and accurately in this paper , such as the neglect of graph structures and the constraints posed by limited labeled data.
+ Solid theoretical foundation for the proposed framework was provided for the proposed framework in the generalization error bound. And methods are proven and discussed mathematically.
+  The introduction of dual structure-aware adapters is a noteworthy contribution, as it allows for the adaptive integration of task-related structural information from both homogeneous and heterogeneous graphs.
+ Experiments are comprehensive enough to demonstrate the effectiveness and generalization of the proposed method across different tasks add credibility to the claims made.

Weaknesses:
+ The paper could benefit from clearer explanations of key concepts, particularly around the implementation of the dual structure-aware adapters and the label-propagated contrastive loss.
+ While the empirical results show improvements over existing methods, a more detailed comparative analysis with specific baselines would enhance the reader's understanding of the method's relative performance.
+ The paper would be stronger with a more explicit discussion of the limitations of the proposed method, particularly in relation to scenarios with highly variable graph structures or the quality of unlabeled data.
+  It might be recommendable to talk about future work considerations to make background and trend of the overall research clear.

**Strengths:**

See Summary

**Weaknesses:**

See Summary

**Questions:**

See Summary

---

> ### Author Response · Authors · 2024-11-22
> **Response to Reviewer J7pK**
>
> Thanks for the positive comments on the novelty, theoretical analysis, and experimental results of our method. We are so encouraged and will try our best to address the concerns one by one.
>
> > **Q1.** The paper could benefit from clearer explanations of key concepts, particularly around the implementation of the dual structure-aware adapters and the label-propagated contrastive loss.
>
> **A1**. Thanks for your suggestion. We summarize the implementation of the dual structure-aware adapters and the label-propagated contrastive loss as follows.
>
> First, the dual structure-aware adapters are designed to model both node features as well as homogeneous and heterogeneous graph structures. Specifically, the homogeneous adapter includes two parts (i.e.,  feature and graph structure tuning). In the feature tuning part, we employ the two-layer MLP $f_\delta: \mathbb{R}^{N \times d}\to \mathbb{R}^{N \times d'}$ to obtain the mapped representations $\mathbf{F}$ of the frozen representations
> $\tilde{\mathbf{H}}$. In the graph structure tuning part, we first employ another MLP $f_\vartheta: \mathbb{R}^{N \times d}\to \mathbb{R}^{N \times d''}$ to obtain new representations of the frozen representations $\tilde{\mathbf{H}}$. After that, we calculate the similarity weight $\tilde{\mathbf{a}}_{i,j}$ between new representations of nodes $v_i$ and $v_j$ from the same node type to tune the homogeneous graph structure adaptively. Finally, we conduct the message-passing based on the tuned features and graph structures. The heterogeneous adapter shares a similar process.
>
> Second, the label-propagated contrastive loss is designed to bridge the gap between different pre-trained models and downstream tasks as well as extend the potential labeled data. Specifically, we first obtain the propagated labels for unlabeled nodes based on the learned homogeneous graph structure $\mathbf{A}$ and the given node labels. Then we employ a projection $g_\rho: \mathbb{R}^{N \times d'} \to \mathbb{R}^{N \times c}$ to map node representations $\mathbf{Z}$, resulting in the prediction matrix $\mathbf{P}$ of all nodes, where $c$ denotes the number of classes. We then obtain the class subgraph predictions by averaging the prediction vectors of nodes with the same original label.  After that, we propose a contrastive loss based on the subgraph similarity to incorporate supervision signals by enforcing the node prediction $\mathbf{p}_i$ close to its class-subgraph prediction while far away from different class-subgraph predictions.
>
> > **Q2.** While the empirical results show improvements over existing methods, a more detailed comparative analysis with specific baselines would enhance the reader's understanding of the method's relative performance.
>
> **A2.** Thanks for the suggestion. The proposed method is designed to improve different pre-trained HGNNs. In our experiments, we implement our HG-Adapter on three pre-trained HGNNs (i.e., HDMI, HeCo, HERO) and obtains significant relative improvements on these baselines. For instance, the proposed method on average, improves by 1.8%, 1.3%, 1.3%, compared to HDMI, HeCo, HERO, respectively, on all heterogeneous graph datasets. This further verify the effectiveness of the proposed method on improving different pre-trained HGNNs.
>
> > **Q3**. The paper would be stronger with a more explicit discussion of the limitations of the proposed method, particularly in relation to scenarios with highly variable graph structures or the quality of unlabeled data.
>
> **A3**. Thanks for the suggestion. In our original submission, we discussed the limitations of the proposed method in Appendix F. In the revision, we further discussed the limitations in relation to scenarios with highly variable graph structures or the quality of unlabeled data.
>
> > **Q4.** It might be recommendable to talk about future work considerations to make background and trend of the overall research clear.
>
> **A4.** Thanks for the suggestion. In our original submission, we discussed the future work in Appendix F.

---

> > ### Comment · Reviewer_J7pK · 2024-11-22
> > **To authors**
> >
> > I think all the questions have been solved, and I would like to update my score to 8.

---

### Official Review · Reviewer_NAje · 2024-11-09

**Soundness:** 2
**Presentation:** 3
**Contribution:** 2
**Rating:** 6
**Confidence:** 3

**Summary:**

In this paper, a unified pre-trained and prompt tuning framework is proposed by combining two new adapters with potential labeled data extension to improve the generalization of pre-trained heterogeneous graph neural networks. In the proposed method, dual structure-aware adapters are adopted to fit task-related homogeneous and heterogeneous structural information. Meanwhile, three losses are designed, including a label-propagated contrastive loss and two self-supervised losses. The ablation studies show that each component is very essential in the proposed method.

**Strengths:**

In the proposed method, dual structure-aware adapters are designed to capture task-related homogeneous and heterogeneous structural information. A label-propagated contrastive loss and two self-supervised losses are proposed to achieve the potential labeled data extension. Meanwhile, a theoretical analysis also has been given to verify the effectiveness of the proposed method.

**Weaknesses:**

The motivation should be further explained with some toy examples. In the analysis part of challenges, although three limitations are presented, how to demonstrate that their drawbacks really exist in the real-world applications.

**Questions:**

1)	In the main contributions, the authors has highlighted that “it is the first dedicated attempt to design a unified “pre-train, adapter-tuning” paradigm to improve different pre-trained HGNN models”. While HGPrompt (Yu et al., 2024a) is designed with both pre-training and the prompt-tuning. Hence, I wonder whether the first effort is suitable.
2)	In the experiments, for the method HERO, the improvements are not satisfactory except on Aminer. Hence, I wonder the complexity between HG-Adapter and HetGPT.
3)	Although the proposed method uses the self-supervised technique to improve the confidence of propagated labels, I do not think it can guarantee the accuracy of the propagated labels. Hence, I wonder if the propagated labels are too noisy, whether the proposed method can obtain satisfactory performance.

---

> ### Author Response · Authors · 2024-11-22
> **Response to Reviewer NAje (part 1)**
>
> Thanks for the positive comments on our method and theoretical results. We are so encouraged and will try our best to address the concerns one by one.
>
> > **Q1.** The motivation should be further explained with some toy examples. In the analysis part of challenges, although three limitations are presented, how to demonstrate that their drawbacks really exist in the real-world applications.
>
> **A1**. Thanks for your suggestion. In this paper, we point out that existing methods only focus on node features while ignoring graph structures, thereby increasing the training error. Moreover, existing methods may be constrained by the limited labeled data during the prompt-tuning stage, leading to a large generalization gap. To explain the above motivation, we sample a few nodes from the ACM dataset and construct a toy example. We then implement adapter-tuning and label propagation on the toy example, and visualize it in Figure 3 in Appendix E.
>
> From Figure 3, we have the observations as follows. **First**, if we only tune the node features and ignore tuning the graph structures, some nodes in the training set may be misclassified, thereby increasing the training error. For example, for node 11, it will aggregate much information from the nodes (i.e., nodes 10, 12, 13) of another class after the message-passing with the original graph structures. Therefore, this may cause nodes to confuse their own class information, thus increasing the training error. In contrast, if we tune both the node features and the graph structures, the misclassified node 11 can be corrected by re-weighting the edge weight. As a result, the proposed method  decreases the training error thus improve the model generalization. **Second**, compared with unlabeled nodes, the ratio of labeled nodes is very small, result in a large generalization gap between the training error and the test error. However, after the label propagation in Figure 3, the number of labeled nodes increases greatly. As a result, the proposed method decreases the generalization gap thus improve the model generalization.
>
> In addition, in the original submission, to further verify the motivation of the proposed method, we conducted the ablation study by removing the  graph structure-tuning module and the labeled data extension module, respectively, and reported the corresponding training error and generalization gap in Figure 4 and Figure 5 in Appendix E.  Obviously, the proposed method with structure tuning obtains consistently lower training error than the method without structure tuning. Moreover, the proposed method with the potential labeled data extension consistently achieves a smaller generalization gap than the method without label extension. Therefore, the  motivation of the graph structure-tuning and the labeled data extension is further verified.
>
> For the presented three limitations, their drawbacks widely exist in the real-world applications.
>
> **First**, existing works lack a unified theoretical framework. As a result, those works for real-world applications can only rely on heuristically designed prompts, which may require many experts to just and repeatedly try. In addition, when applied to new applications, the prompt framework may need to be redesigned based on previous experience.
>
> **Second**, existing works generally ignore tuning the graph structure. However, the noise of graph structure is common in real-world applications, such as connections between users in unrelated fields, causes misclassifications. For instance, on platforms like LinkedIn, users might connect across industries, which can introduce noise when trying to classify users into professional clusters.
>
> **Third**, existing works are generally constrained by the limited labeled data. This issue is also very common in real-world applications. For example, platforms like Facebook and Twitter face restrictions in collecting detailed user labels, leading to sparse labeled datasets for training models. In addition, in medical research, acquiring labeled patient data is limited by privacy regulations, which is a common issue in diagnostics and personalized medicine.
>
> > **Q2.** In the main contributions, the authors has highlighted that “it is the first dedicated attempt to design a unified “pre-train, adapter-tuning” paradigm to improve different pre-trained HGNN models”. While HGPrompt (Yu et al., 2024a) is designed with both pre-training and the prompt-tuning. Hence, I wonder whether the first effort is suitable.
>
> **A2**. As the reviewer mentioned, HGPrompt proposes  “**pre-train, prompt-tuning**” paradigm for the heterogeneous graph by designing a learnable prompt that directly appends to (or modifies) the model input. In contrast, the proposed method makes the first attempt to design a “**pre-train, adapter-tuning**” paradigm to tune both node features and graph structures in the heterogeneous graph by lightweight neural networks.

---

> ### Author Response · Authors · 2024-11-22
> **Response to Reviewer NAje (part 2)**
>
> > **Q3.** In the experiments, for the method HERO, the improvements are not satisfactory except on Aminer. Hence, I wonder the complexity between HG-Adapter and HetGPT.
>
> **A3**. The proposed HG-Adapter and the comparison method HetGPT are both designed to tune different pre-trained HGNNs. It is worth noting that certain HGNNs (e.g., HERO) already achieve relatively high performance on the given tasks. As a result, further improvements in these models can appear marginal due to the strong existing baselines. However, while HetGPT shows limited gains on such high-performing models, our proposed method consistently achieves more significant improvements, demonstrating its effectiveness when applied to these strong baselines.
>
> In the revision, we further evaluated the proposed method on the biomedical heterogeneous graph dataset (i.e., HBN-B) and the large-scale heterogeneous graph dataset (i.e., Ogbn-mag) and report the results in Table 5 in Appendix E. The proposed method on average, improves by 1.3% and 3.0%, compared to prompt-tuning-based methods (i.e., HGPrompt and HetGPT) on the HBN-B and Ogbn-mag datasets, respectively. This further verifies the effectiveness of the proposed method on datasets from different domains and large-scale datasets.
>
> In addition, according to the suggestion, we list the time complexity between HG-Adapter and HetGPT as follows.
>
> Complexity of HetGPT: HetGPT consists of four parts, i.e., virtual class prompt, heterogeneous feature prompt, multi-view neighborhood aggregation, and prompt-based learning and inference. We analyze the time complexity of each part as follows.
> First, the time complexity of the virtual class prompt is $\mathcal{O}(n_c)$, where $n_c$ indicates the number of labeled nodes. Second, the time complexity of the heterogeneous feature prompt is $\mathcal{O}(nb)$, where $n$ and $b$ indicate the number of all nodes and the size of independent basis vectors, respectively. Third, the time complexity of the multi-view neighborhood aggregation is $\mathcal{O}(nm + np)$, where  $m$ and $p$ indicate the number of node types and the number of meta-paths, respectively. Fourth, the time complexity of the prompt-based learning and inference is $\mathcal{O}(ncd + c^2d)$, where $c$ and $d$ indicate the number of classes and the number of prompt dimensions, respectively. Therefore, the overall time complexity of HetGPT is $\mathcal{O}(n_c + n(b + m + p + cd) + c^2d)$, where $b + m + p + cd$ is usually much smaller than $n$.
>
> Complexity of the proposed HG-Adapter: HG-Adapter consists of two prats, i.e., dual structure-aware adapters and potential labeled data extension. We analyze the time complexity of each part as follows.
> First, the time complexity of the dual structure-aware adapters is $\mathcal{O}(nkd + n|\mathcal{R}|)$, where $n$, $k$, $d$, and $|\mathcal{R}|$ indicate the number of nodes, the number of neighbors of each node, the number of representation dimensions, and the number of edge types, respectively. Second, the time complexity of the potential labeled data extension is $\mathcal{O}(nkc + nc^2 + nkf)$, where $c$ and $f$ indicate the number of classes and dimensions of node features, respectively. Therefore, the overall time complexity of the proposed HG-Adapter is $\mathcal{O}(n(kd + |\mathcal{R}| + kc + c^2 + kf))$, where $kd + |\mathcal{R}| + kc + c^2 + kf$ is usually much smaller than $n$.
>
> Based on the above analysis, HG-Adapter always obtains more significant improvements on baselines than HetGPT and shows comparative time complexity with HetGPT (i.e., both scaled linearly with the sample size).
>
> [1] Heterogeneous Graph Attention Network for Drug-Target Interaction Prediction. In CIKM 2022.
>
> [2] Open Graph Benchmark: Datasets for Machine Learning on Graphs. In NeurIPS 2020.

---

> ### Author Response · Authors · 2024-11-22
> **Response to Reviewer NAje (part 3)**
>
> > **Q4.** Although the proposed method uses the self-supervised technique to improve the confidence of propagated labels, I do not think it can guarantee the accuracy of the propagated labels. Hence, I wonder if the propagated labels are too noisy, whether the proposed method can obtain satisfactory performance.
>
> **A4.**  The accuracy of the propagated labels relies on the quality of the learned homogeneous graph structure $\mathbf{A}$. That is, if the learned graph structures possess a high homophily ratio (the ratio of edges that connect nodes within the same class), the propagated labels are more likely to be accurate. Conversely, if the graph structure has a low homophily ratio, the propagated labels will be noisy.
>
> Therefore, we design the self-supervised losses not only to improve the confidence of propagated labels but also to optimize the homogeneous graph structure and improve its homophily ratio. Specifically,  we design the feature reconstruction loss to enforce the reconstructed node features after message-passing to be aligned with the original node features. This requires that message-passing occurs only among nodes with similar node features. As a result, the reconstruction loss encourages the graph structure $\mathbf{A}$ to connect nodes within the same class while disconnecting nodes from different classes as much as possible to improve its homophily ratio.
>
> In the original submission, to verify the quality of the learned homophily graph structure $\mathbf{A}$, we reported the homophily ratios of the homogeneous graph structure $\mathbf{A}$ learned by HERO+HG-Adapter
> on four datasets in Figure 2. Obviously, the proposed method obtains a relatively high homophily ratio on four datasets, especially on the Yelp and Aminer datasets ($>$ 80%). This indicates that the proposed self-supervised loss indeed optimizes the graph structure $\mathbf{A}$ to improve its homophily ratio as well as avoid the propagated labels that are too noisy.

---

> ### Author Response · Authors · 2024-11-28
>
> Dear Reviewer NAje,
>
> Thank you once again for your valuable and constructive feedback on our submission. As the discussion phase is nearing its conclusion, we want to kindly confirm if we have adequately addressed all your concerns. Please do not hesitate to let us know if there are any remaining questions or points that require further clarification.
>
> Sincerely,
>
> Authors of Submission3354

---

> ### Author Response · Authors · 2024-12-01
> **Gentle reminder for Reviewer NAje**
>
> Dear Reviewer NAje,
>
> As the rebuttal is coming to a close, we would like to provide a gentle reminder that we have posted a response to your comments. May we please check if our responses have addressed your concerns and improved your evaluation of our paper? We are happy to provide further clarifications to address any other concerns that you may still have before the end of the rebuttal.
>
> Sincerely,
>
> Authors of Submission3354

---

> ### Author Response · Authors · 2024-12-03
>
> Dear Reviewer NAje,
>
> We sincerely appreciate your insightful feedback and the time you have dedicated to reviewing our submission. Given that the rebuttal deadline is approaching, we kindly inquire whether our responses and revisions sufficiently address your concerns. If there are any remaining issues or suggestions, we are fully prepared to make further clarification promptly.
>
> We deeply appreciate the reviewer's dedication throughout this process and eagerly anticipate your further feedback.
>
> Sincerely,
>
> Authors of Submission3354

---

### Official Review · Reviewer_hjNU · 2024-11-12

**Soundness:** 3
**Presentation:** 3
**Contribution:** 3
**Rating:** 6
**Confidence:** 2

**Summary:**

A dual-adapter approach has been introduced to graph prompt-tuning methods. The key insight of this work is to leverage dual adapters to capture both node features and graph structures to realize prompt tuning for pre-trained HGNN models. Experimental results on four benchmark datasets were provided in terms of two validation metrics.

**Strengths:**

- A new dual-adapter approach, applied to different pre-trained HGNN models, has been developed to enrich the current graph prompt tuning methods.
- Two self-supervised graph learning loss functions have been designed to realize label augmentation, showing a clear improvement in the ablation study given in `Table 2`.
- The paper provides two orthogonal solutions to improve the generalization error bound, i.e., 1) improving prompt flexibility/complexity by dual adapters and 2) improving data efficiency by designing self-supervised label augmentation.

**Weaknesses:**

- **Limited Technical Novelty**: The design of dual adapters and self-supervised losses builds on top of several well-established technologies. Thus, the technical contribution of this work is relatively limited since no new loss functions or adapter architectures have been developed.
- **Unclear Insights for HGNN**: It remains unclear what new insights or hypotheses have been introduced to HGNNs by this work. It seems common to improve the generalization bound by increasing parameters and labels. Are there any specific designs or findings for prompt-tuning of HGNNs? The theoretical results in `Theorem 2.3` and `Theorem 2.4` are relatively weak, where the former only proves the existence of $|\bar{P}_M|$ without giving guidance on how to tune or find this optimal size, while the latter builds on *a very strong assumption* that $|P_A|$ is expected to be closer to $|\bar{P}_M|$ without showing evidence.
- **Marginal Improvement**: As shown in `Table 1`, the improvement of the proposed HG-Adapter over existing graph prompt tuning methods, e.g., HGPrompt and HetGPT, is quite marginal and generally less than 1% in three of four datasets. Per the provided theoretical results, can we further improve the performance by increasing the size of dual adapters? It would also be helpful to provide an abolition study on each adapter and give a parameter analysis in terms of adapter size.

**Questions:**

- Can the authors more clearly articulate the novel aspects of their approach, particularly in combining dual adapters with prompt tuning for heterogeneous graphs?
- Are there any specific insights or hypotheses about prompt-tuning for HGNNs that motivated the proposed approach? It would also be helpful to explain how to find or approximate the optimal parameter size $|\bar{P}_M|$ and provide empirical evidence or further justification to demonstrate that $|P_A|$ is closer to $|\bar{P}_M|$.
- Can the performance be further improved by varying the size of the dual adapters? An ablation study showing the impact of each adapter individually is also expected.

---

> ### Author Response · Authors · 2024-11-22
> **Response to Reviewer hjNU (part 1)**
>
> Thanks for the constructive comments on our method. We are so encouraged and will try our best to address the concerns one by one.
>
> > **Q1.** Limited Technical Novelty: The design of dual adapters and self-supervised losses builds on top of several well-established technologies. Thus, the technical contribution of this work is relatively limited since no new loss functions or adapter architectures have been developed.
>
> **A1.**  The proposed method does not simply transfer well-established technologies in existing works. In contrast, the dual adapters and self-supervised losses in this paper are both designed specifically for the heterogeneous graph. Compared to these adapters and self-supervised losses in existing works, the technical novelty of the proposed method can be summarized as follows.
>
> First, adapters in existing works [1, 2] are generally designed to tune the sample features by adding lightweight neural networks. However, these adapters may not easily transferred to the heterogeneous graph due to they cannot deal with the complex structures in the heterogeneous graph. To solve this issue, this work makes the first attempt to design dual structure-aware adapters to tune node features as well as homogeneous and heterogeneous structures.
>
> Second, self-supervised losses in existing works [3, 4] are generally designed to extract the invariant information between the original graph view and the augmented graph view, thus obtaining discriminative representations. However, these self-supervised losses may not directly transferred to our framework due to the need to optimize the graph structures in dual adapters. To do this, this work designs the feature reconstruction loss and the margin loss to optimize graph structures as well as achieve the potential labeled data extension.
>
> [1] Lora: Low-Rank Adaptation of Large Language Models. In ICLR 2021.
>
> [2] Semantically-Shifted Incremental Adapter-Tuning is A Continual ViTransformer. In CVPR 2024.
>
> [3] Contrastive Multi-View Representation Learning on Graphs. In ICML 2020.
>
> [4] A Survey on Self-Supervised Learning: Algorithms, Applications, and Future Trends. TPAMI 2024.
>
> > **Q2.** Unclear Insights for HGNN: It remains unclear what new insights or hypotheses have been introduced to HGNNs by this work. It seems common to improve the generalization bound by increasing parameters and labels. Are there any specific designs or findings for prompt-tuning of HGNNs? The theoretical results in Theorem 2.3 and Theorem 2.4 are relatively weak, where the former only proves the existence of $|\bar{P}_M|$ without giving guidance on how to tune or find this optimal size, while the latter builds on a very strong assumption that $|{P}_A|$ is expected to be closer to $|\bar{P}_M|$ without showing evidence.
>
> **A2.** According to Theorem 2.3, the generalization bound cannot improved by simply increasing parameters due to the following reasons. First,  if the number of training samples is fixed and the parameter size increases, the upper bound of the test error (i.e., $\mathcal{U}(\mathcal{E}_M)$) will first decrease until it reaches the lowest point (i.e., $\min (\mathcal{U}(\mathcal{E}_M))$) and then start to increase. That is, the upper bound of the test error exhibits the U-shaped pattern with the increase of the parameters.
> Second,  the upper bound of the test error will further decrease with the increase of the training samples. Based on the above observation, we have the insights and findings for the prompt-tuning of HGNNs as follows.
>
> **First**, according to the first observation, to improve the generalization of pre-trained HGNNs, we can enable the parameters size approaches $|\bar{P}_M|$ to approach the lowest upper bound of the test error (i.e., $\min (\mathcal{U}(\mathcal{E}_M))$).
> As the reviewer mentioned, we prove the existence of $|\bar{P}_M|$ in Theorem 2.3, but we cannot directly find the optimal size. Actually, Theorem 2.3 indicates that the upper bound of the test error consists of two parts, i.e., the training error $\hat{\mathcal{E}}_M$ of the model in the prompt-tuning stage, and the generalization gap bound. Therefore, although we cannot directly find the optimal parameter size via Theorem 2.3, it provides guidance on how to approach the lowest upper bound of the test error. That is, if the number of training samples $n_M$ is fixed, we can better fit the input data with few parameters to decrease the training error, thus decreasing the upper bound of the test error and approaching the optimal parameters $|\bar{P}_M|$.

---

> ### Author Response · Authors · 2024-11-22
> **Response to Reviewer hjNU (part 2)**
>
> To do this, we point out that existing prompt-tuning-based methods always focus on node features while ignoring graph structures. As a result, the parameters in existing methods may be insufficient to fit the input data (node feature and graph structures) effectively, leading to the increased training error. Therefore, we design dual structure-aware adapters with few additional parameters to model both node features as well as homogeneous and heterogeneous graph structures, thus fitting the input data better to decrease the training error. According to Theorem 2.3, the upper bound of the test error will also decrease. Therefore, this enables our parameters $|{P}_A|$  closer to the optimal parameters $|\bar{P}_M|$.
>
> In the revision, to verify that $|{P}_A|$ is indeed closer to $|\bar{P}_M|$, we fix the number of training samples, and then implement several variant methods (i.e., adapter-tuning on both node features and graph structures, adapter-tuning on only node features, and prompt-tuning on only node features), and report the results in Figure 6 in Appendix E. Obviously, when the number of training samples is fixed, the proposed adapter-tuning on node features and graph structures always obtains lower test error than the prompt-tuning and adapter-tuning on only node features. As a result, we can obtain that the parameters $|{P}_A|$ of the proposed adapter-tuning is indeed closer to $|\bar{P}_M|$ than existing prompt-tuning-based methods. Therefore, it is actually a mild assumption that $|P_A|$ is expected to be closer to $|\bar{P}_M|$ in Theorem 2.4.
>
> **Second**, according to the second observation, to improve the generalization of pre-trained HGNNs, we can increase the number of training samples to further decrease the generalization gap of pre-trained HGNNs. However, obtaining a large number of labeled data is challenging and costly in real scenarios. To solve this issue, in this paper, we design a label-propagated contrastive loss and two self-supervised losses, extending all unlabeled nodes as the potential labeled data to further improve the model's generalization ability.
>
> In our original submission, to verify the effectiveness of the potential labeled data extension, we investigated the generalization gap of the proposed method with and without the label extension, and report the results in Figure 5 in Appendix E. From Figure 5, we can find that the proposed method with the labeled data extension consistently achieves a smaller generalization gap than the method without the labeled data extension. This is reasonable because the label extension increases the number of training samples potentially, thus decreasing the generalization gap bound and further decreasing the generalization error bound of existing methods.

---

> ### Author Response · Authors · 2024-11-22
> **Response to Reviewer hjNU (part 3)**
>
> > **Q3.** Marginal Improvement: As shown in Table 1, the improvement of the proposed HG-Adapter over existing graph prompt tuning methods, e.g., HGPrompt and HetGPT, is quite marginal and generally less than 1% in three of four datasets. Per the provided theoretical results, can we further improve the performance by increasing the size of dual adapters? It would also be helpful to provide an abolition study on each adapter and give a parameter analysis in terms of adapter size.
>
> **A3**. The proposed HG-Adapter and the comparison method HetGPT are both designed to tune different pre-trained HGNNs. It is worth noting that certain pre-trained HGNNs (e.g., HERO) already achieve relatively high performance on the given tasks. As a result, further improvements in these models can appear marginal due to the strong existing baselines. However, while HetGPT shows limited gains on such high-performing models, our proposed method consistently achieves more significant improvements, demonstrating its effectiveness when applied to these strong baselines.
>
> In the revision, we further evaluate the proposed method on the biomedical heterogeneous graph dataset (i.e., HBN-B [5]) and the large-scale heterogeneous graph dataset (i.e., Ogbn-mag [6]), and report the results in Table 5 in Appendix E. The proposed method on average, improves by 1.3% and 3.0%, compared to prompt-tuning-based methods HGPrompt and HetGPT, respectively, on the HBN-B and Ogbn-mag datasets. This further verifies the effectiveness of the proposed method on datasets from different domains and large-scale datasets.
>
> In addition, based on the theoretical results in Theorem 2.3, the upper bound of the test error exhibits a U-shaped pattern, where it initially decreases and then increases as the number of parameters increases. Therefore, we may not further improve the performance by simply increasing the size of dual adapters. To verify it, we conduct the ablation study by varying the size of each adapter, and report the results in Figure 7 in Appendix E. From Figure 7, we can find that as the adapter size increases, the performance of the model may first increase, and then decrease when the size is too large. This is consistent with our theoretical results above, i.e., as the parameter size increases, the upper bound of the test error decreases first and then increases. Correspondingly, the performance of the model may increase first and then decrease. This actually verifies the motivation of our method, i.e., instead of directly increasing the size of parameters of the model, we aim to use few parameters to better fit the input data (i.e., node features and graph structures), thereby reducing training error.
>
> Moreover, in our original submission, we conducted the ablation study to verify the effectiveness of each adapter by individually removing homogeneous and heterogeneous adapters, and reported the results in Table 6 in Appendix E.  From Table 6, the proposed method with dual adapters obtains superior performance than the variant methods without either the homogeneous or heterogeneous adapter. Therefore, the effectiveness of each adapter is verified.
>
> [5] Heterogeneous Graph Attention Network for Drug-Target Interaction Prediction. In CIKM 2022.
>
> [6] Open Graph Benchmark: Datasets for Machine Learning on Graphs. In NeurIPS 2020.
>
> > **Q4.**  Can the authors more clearly articulate the novel aspects of their approach, particularly in combining dual adapters with prompt tuning for heterogeneous graphs?
>
> **A4.**  We summarize the technical novelty of the proposed method in **A1**.
>
> > **Q5.** Are there any specific insights or hypotheses about prompt-tuning for HGNNs that motivated the proposed approach? It would also be helpful to explain how to find or approximate the optimal parameter size $|\bar{P}_M|$ and provide empirical evidence or further justification to demonstrate that $|{P}_A|$ is closer to $|\bar{P}_M|$.
>
> **A5**.  We provide the insights about prompt-tuning for HGNNs, explanations on how to approach the optimal parameter size, and empirical evidence in **A2**.
>
> > **Q6**. Can the performance be further improved by varying the size of the dual adapters? An ablation study showing the impact of each adapter individually is also expected.
>
> **A6**. No, the performance may not be further improved by simply varying the size of the dual adapters. To verify it, in the revision, we further conduct the ablation study by varying the size of each adapter, and report the results in Figure 7 in Appendix E.
> Moreover, in our original submission, we conducted the ablation study and verified the impact of each adapter by individually removing homogeneous and heterogeneous adapters, and reported the results in Table 6 in Appendix E. Details can be found in **A3**.

---

> ### Author Response · Authors · 2024-11-28
>
> Dear Reviewer hjNU,
>
> Thank you once again for your valuable and constructive feedback on our submission. As the discussion phase is nearing its conclusion, we want to kindly confirm if we have adequately addressed all your concerns. Please do not hesitate to let us know if there are any remaining questions or points that require further clarification.
>
> Sincerely,
>
> Authors of Submission3354

---

> ### Author Response · Authors · 2024-12-01
> **Gentle reminder for Reviewer hjNU**
>
> Dear Reviewer hjNU,
>
> As the rebuttal is coming to a close, we would like to provide a gentle reminder that we have posted a response to your comments. May we please check if our responses have addressed your concerns and improved your evaluation of our paper? We are happy to provide further clarifications to address any other concerns that you may still have before the end of the rebuttal.
>
> Sincerely,
>
> Authors of Submission3354

---

> > ### Comment · Reviewer_hjNU · 2024-12-01
> > **Post-Rebuttal Feedback**
> >
> > Thanks for the clarification. The reviewer appreciates the new results and more explanations on `Theorem 2.3/2.4`. Particularly, the experiment on increasing adapter size supports the proposed theoretical results well. Thus, the reviewer will increase the rating.

---

### Author Response · Authors · 2024-11-22
**Summary of Author Response to All the Reviewers**

We would like to thank all the reviewers for their insightful comments. We would like to appreciate all the reviewers for their insightful comments. We revised the manuscript based on the constructive feedback and suggestions from the reviewers. We marked the contents that already existed in the original submission (but may be missed by reviewers) in red, and those revised or newly added contents in blue in the revision. Our key responses are summarized as follows:

**> Additional explanation.**

As Reviewer hjNU suggested, we summarized the technical novelty of the proposed method compared to existing works. In addition, we provided more insights and findings for the prompt-tuning of HGNNs. Moreover, we discussed the relationships between the model performance and the size of dual adapters.

As Reviewer NAje suggested, we explained our motivation with the toy example. In addition, we discussed the discussed the presented limitations and their widespread existence in real-world applications. In addition, we analyzed the complexity of the proposed method and the prompt-tuning-based HetGPT and to verify the efficiency of the proposed method. Moreover, we explain the relationship between the accuracy of propagated labels and the proposed self-supervised losses.

As Reviewer J7pK suggested, we explained the implementation of the dual structure-aware adapters and the label-propagated contrastive loss in this paper to make them clear. In addition, we analyzed the improvements of the proposed method, compared to specific baselines. Moreover, we discussed the limitations and future works related to this research.

As Reviewer PQvi suggested, we analyzed the complexity of the proposed method to verify the efficiency and scalability of the proposed method.

**> Additional experimental results.**

As the Reviewer hjNU suggested, we conducted the ablation study and comparison experiments to verify that $|P_A|$ is indeed closer to the optimal $|\bar{P}_M|$. Moreover, we conducted the ablation study to show the relationship between the model performance and the adapter size.

As Reviewer NAje suggested, we constructed the toy example and illustrated our motivation with it.

As Reviewer PQvi suggested, we evaluated the proposed method on datasets from different domains and large-scale dataset to demonstrate the effectiveness and scalability of the proposed method.

**> Summary.**

We thank all the reviewers again for the detailed and constructive review. We are pleased to see the reviewers' acknowledgment of the contribution of the proposed method. Most of the concerns are raised to part of the unclear expressions, and experiments. We hope our explanation, and additional experimental results in the rebuttal could address all of your concerns. Please let us know if you have any  questions or concerns.

---

### Author Response · Authors · 2024-12-04
**Thank the Reviewers for their constructive engagement**

We would like to express our sincere gratitude to the reviewers for their thoughtful feedback and constructive engagement throughout the rebuttal process.

We are pleased that the concerns raised by most reviewers have been effectively addressed, as acknowledged in the rebuttal. The clarifications and additional results we provided have improved the manuscript, and we appreciate the reviewers' recognition of these efforts.

We are also encouraged by the reviewers' positive comments regarding the strengths of our work. Specifically, we are pleased that they appreciate the work's motivation (Reviewers J7pK, PQvi),  novelty (Reviewers J7pK, PQvi), theoretical contribution (Reviewers hjNU, NAje, J7pK, PQvi), and extensive experiments (Reviewers hjNU, J7pK, PQvi). These affirmations further validate the significance of our work.

Thank you again for your time, insightful comments, and the effort you dedicated to improving our work.

Sincerely,

Authors of Submission3354

---

### Meta-Review · Area_Chair_yD3L · 2024-12-11

**Metareview:**

This paper claims that existing prompt-tuning-based works face two limitations: (i) the model may be insufficient to fit the graph structures well as they are generally ignored in the prompt-tuning stage, increasing the training error to decrease the generalization ability; and (ii) the model may suffer from the limited labeled data during the prompt-tuning stage, leading to a large generalization gap between the training error and the test error to further affect the model generalization. To alleviate the above limitations, this paper first derives the generalization error bound for existing prompt-tuning-based methods, and then propose a unified framework that combines two new adapters with potential labeled data extension to improve the generalization of pre-trained HGNN models. Specifically, the authors design dual structure-aware adapters to adaptively fit task-related homogeneous and heterogeneous structural information. The authors further design a label-propagated contrastive loss and two self-supervised losses to optimize dual adapters and incorporate unlabeled nodes as potential labeled data. Theoretical analysis indicates that the proposed method achieves a lower generalization error bound than existing methods, thus obtaining superior generalization ability.


The idea and main findings of this paper are interesting. The theoretical studies in this paper are also sufficient.  The experimental results also demonstrate the effectiveness of the proposed method.

**Additional Comments On Reviewer Discussion:**

After rebuttal, the reviewers' previous concerns have been well addressed by the authors, and all four reviewers show positive scores to this paper. Some of the reviewers also raised the scores. Therefore, I recommend and acceptance.

---

### Decision · Program_Chairs · 2025-01-22

Accept (Poster)